# Resetting the Optimizer in Deep RL:
# An Empirical Study

**Kavosh Asadi**
Amazon

**Rasool Fakoor**
Amazon

**Shoham Sabach**
Amazon & Technion

## Abstract

We focus on the task of approximating the optimal value function in deep reinforcement learning. This iterative process is comprised of solving a sequence of optimization problems where the loss function changes per iteration. The common approach to solving this sequence of problems is to employ modern variants of the stochastic gradient descent algorithm such as Adam. These optimizers maintain their own internal parameters such as estimates of the first-order and the second-order moments of the gradient, and update them over time. Therefore, information obtained in previous iterations is used to solve the optimization problem in the current iteration. We demonstrate that this can contaminate the moment estimates because the optimization landscape can change arbitrarily from one iteration to the next one. To hedge against this negative effect, a simple idea is to reset the internal parameters of the optimizer when starting a new iteration. We empirically investigate this resetting idea by employing various optimizers in conjunction with the Rainbow algorithm. We demonstrate that this simple modification significantly improves the performance of deep RL on the Atari benchmark.

## 1 Introduction

Value-function optimization lies at the epicenter of large-scale deep reinforcement learning (RL). In this context, the deep RL agent is equipped with a parameterized neural network that, in conjunction with large-scale optimization, is employed to find an approximation of the optimal value function. The standard practice is to initialize the optimizer only once and at the beginning of training [1]. With each step of gradient computation, the optimizer updates its internal parameters, such as the gradient's moment estimates [2], and uses these internal parameters to update the network.

A unique ingredient in value-function optimization is a technique known as forward bootstrapping [3]. In typical regression problems, the loss function is a measure of the discrepancy between a fixed target and the agent's current approximation. This stands in contrast to value-function optimization with forward bootstrapping. In this setting, the goal is to minimize the discrepancy between the agent's current approximation and a second approximation obtained by performing one or multiple steps of look ahead. The regression target thus depends on the RL agent's own approximation which changes continually over the span of learning.

We first argue that in the presence of forward bootstrapping the value-function optimization process can best be thought of as solving a sequence of optimization problems where the loss function changes per iteration. We demonstrate that in this case the loss function being minimized is comprised of two inputs: target parameters that remain fixed during each iteration, and optimization (or online) parameters that are adjusted to minimize the loss function during each iteration. In this context, updating the target parameters changes the loss function being minimized in the next iteration. By extension, the first-order and the second-order moments of the gradient can also change arbitrarily because they depend on the landscape of the loss function.

37th Conference on Neural Information Processing Systems (NeurIPS 2023).

Realizing that the RL agent faces a sequence of optimization problems, a natural practice would be to reset the optimizer at the beginning of each new optimization problem. However, in deep RL, modern optimization algorithms are employed often without ever resetting their internal parameters. We question this standard practice, and in particular, similar to Bengio et al. [4] we ask if the internal parameters accumulated by the optimizer in the previous iterations can still be useful in updating the parameters of the neural network in the current iteration. Could it be the case that in most cases relying on gradient computations pertaining to the previous iterations is just contaminating the internal parameters? And ultimately, can this negatively affect the performance of the RL agent? We answer these questions through various experiments.

We then propose a simple modification to existing RL agents where we reset the internal parameters of their optimizer at the beginning of each iteration. Using the deep RL terminology, each time we update the target network, we also reset the internal parameters of the optimizer. We show that this remarkably simple augmentation significantly improves the performance of the competitive Rainbow agent [5] when used in conjunction with various optimizers. Most notably, under the standard Adam optimizer [6], we observe that resetting unleashes the true power of Adam and ultimately results in much better reward performance, suggesting that resetting the optimizer is a more promising choice in the context of value-function optimization.

## 2 RL as a Sequence of Optimization Problems

We argue that many deep RL algorithms can be viewed as iterative optimization algorithms where the loss function being minimized changes per iteration. To this end, we first recall that the DQN algorithm [7] decouples the learning parameters into two sets of parameters: the target parameters $\theta$, and the optimization (or online) parameters $w$ that are adjusted at each step. DQN updates these two parameters in iterations. More specifically, each iteration is comprised of an initial step, $w^{t,0} = \theta^t$, performing multiple updates with a fixed $\theta^t$:

$$w^{t,k+1} \leftarrow w^{t,k} + \alpha \big( r + \gamma \max_{a'} q(s', a'; \theta^t) - q(s, a; w^{t,k}) \big) \nabla_\theta q(s, a; w^{t,k}) \,. \tag{1}$$

The agent then synchronizes the two learning parameters, $\theta^{t+1} \leftarrow w^{t,K}$, prior to moving to the next iteration $t + 1$ where updates are performed using $\theta^{t+1}$. Here $K$ is a hyper-parameter whose value is commonly set to 8000 for DQN and its successors [5].

Observe that the effect of changing the optimization parameters $w$ on $\max_{a'} q(s', a'; \theta^t)$ is ignored during gradient computation despite the fact that an implicit dependence exists due to synchronization. In fact, the update cannot be written as the gradient of any loss function that only takes a single parameter as input [8]. However, this update can be written as the gradient of a function with two input parameters. To illustrate this, we now define:

$$H(\theta, w) = \frac{1}{2} \sum_{\langle s,a,r,s' \rangle \in \mathcal{B}} \big( r + \gamma \max_{a'} q(s', a'; \theta) - q(s, a; w) \big)^2 \,, \tag{2}$$

where $\mathcal{B}$ is the experience replay buffer containing the agent's environmental interactions. Observe that the update (1) could be thought of as performing gradient descent using $\nabla_w H$ on a single sample $\langle s, a, r, s' \rangle$ where $\nabla_w H$ is the partial gradient of $H$ with respect to $w$. Therefore, by viewing the $K$ gradient steps as a rough approximation of exactly minimizing $H$ with respect to the optimization parameters $w$, we can write this iterative process as:

$$\theta^{t+1} \approx \arg\min_w H(\theta^t, w) = \arg\min_w H_t(w) \,. \tag{3}$$

In this optimization perspective [9], what is typically referred to as the online parameter is updated incrementally because the objective function (2) does not generally lend itself into a closed-form solution. Notice also that in practice rather than the quadratic loss, DQN uses the slightly different Huber loss [7]. Moreover, follow-up versions of DQN use different loss functions, such as the Quantile loss [10], sample experience tuples from the buffer non-uniformly [11], use multi-step updates [12, 13], or make additional modifications [5, 14]. That said, the general structure of the algorithm follows the same trend in that it proceeds in iterations, and that the loss function being minimized changes per iteration.

# 3 Revisiting Adam in RL Optimization

So far we have shown that popular deep RL algorithms could be thought of as a sequence of optimization problems where we approximately solve each iteration using first-order optimization algorithms. The most primitive optimizer is the stochastic gradient descent (SGD) algorithm, which simply estimates and follows the descent direction. However, using vanilla SGD is ineffective in the context of deep RL. In contrast, more sophisticated successors of SGD have been used successfully. For example, the original DQN paper [7] used the RMSProp optimizer [15]. Similarly, Rainbow [5] used the Adam optimizer [6], which could roughly be thought of as the momentum-based [16] version of RMSProp. More formally, suppose that our goal is to minimize a certain loss function $J(w)$. Then, starting from $w^0$, after computing a stochastic gradient $g^i \approx J(w^i)$, Adam proceeds by computing a running average of the first-order and the second-order moments of the gradient as follows:

$$m^i \leftarrow \beta_1 m^{i-1} + (1-\beta_1)g^i , \qquad \text{and} \qquad v^i \leftarrow \beta_2 v^{i-1} + (1-\beta_2)(g^i)^2 ,$$

where $(g^i)^2$ applies the square function to $g^i$ element-wise. Notice that in Adam we initialize $m^0 = \mathbf{0}$ and $v^0 = \mathbf{0}$, and so the estimates are biased towards 0 in the first few steps. Define the function $\text{power}(x, y) = x^y$. To remove this bias, a debiasing step is performed in Adam as follows:

$$\widehat{m} \leftarrow m^i/\big(1 - \text{power}(\beta_1, i)\big) , \qquad \text{and} \qquad \widehat{v} \leftarrow v^i/\big(1 - \text{power}(\beta_2, i)\big) ,$$

before finally updating the network parameters: $w^i \leftarrow w^{i-1} - \alpha\widehat{m}/(\sqrt{\widehat{v}} + \epsilon)$ with a small hyper-parameter $\epsilon$ that prevents division by zero. The hyper-parameter $\alpha$ is the learning rate.

As explained before, in deep RL we often minimize a sequence of loss functions, unlike the standard usecase of Adam above where we minimize a fixed loss. We now present the pseudocode of DQN with Adam. Notice that in this pseudocode we do not present the pieces related to the RL agent's environmental interactions to primarily highlight pieces pertaining to how $\theta$ and $w$ are updated. We use $t$ to denote the sequence of loss functions (outer iteration), and we use $k$ to denote each gradient step (inner iteration) in a fixed outer iteration $t$. One can present the code with these two indices, but to simplify the presentation, we use a third index $i$ that is incremented with each gradient step.

---

**Algorithm 1** Pseudocode for DQN with (resetting) Adam

---

**Input:** $\theta^0$, $T$, $K$
**Input:** $\beta_1, \beta_2, \alpha, \epsilon$      ▷ Set Adam's hyper-parameters
$i = 0, m^0 = \mathbf{0}, v^0 = \mathbf{0}$      ▷ Initialize Adam's internal parameters
**for** $t = 0$ **to** $T - 1$ **do**
    $w^{t,0} \leftarrow \theta^t$
    $i = 0, m^0 = \mathbf{0}, v^0 = \mathbf{0}$      ▷ Reset Adam's internal parameters
    **for** $k = 0$ **to** $K - 1$ **do**
        $g^i \leftarrow \nabla H_t(w^{t,k})$
        $i \leftarrow i + 1$
        $m^i \leftarrow \beta_1 m^{i-1} + (1-\beta_1)g^i$ and $v^i \leftarrow \beta_2 v^{i-1} + (1-\beta_2)(g^i)^2$
        $\widehat{m} \leftarrow m^i/\big(1 - \text{power}(\beta_1, i)\big)$ and $\widehat{v} \leftarrow v^i/\big(1 - \text{power}(\beta_2, i)\big)$      ▷ Adam's debiasing step
        $w^{t,k+1} \leftarrow w^{t,k} - \alpha\widehat{m}/(\sqrt{\widehat{v}} + \epsilon)$
    **end for**
    $\theta^{t+1} \leftarrow w^{t,K}$
**end for**
**Return** $\theta^{T-1}$

---

From the pseudocode above, first notice that if the index $i$ is not reset, then the debiasing quantities $1 - \text{power}(\beta_1, i)$ and $1 - \text{power}(\beta_2, i)$ quickly go to 1 and so the debiasing steps will have minimal effect on the overall update, if at all. In absence of resetting, the optimization strategy could then be thought of as initializing the first-order ($m$) and the second-order ($v$) moment estimates at each iteration $t$ by whatever their values were at the end of the previous iteration $t - 1$. This seems like an arbitrary choice, one that deviates from design decisions that were made by Adam [6].

This choice makes some sense if the optimization landscape in the previous iterations is similar to that of the current iteration, but it is not clear that this will always be the case in deep RL. In cases

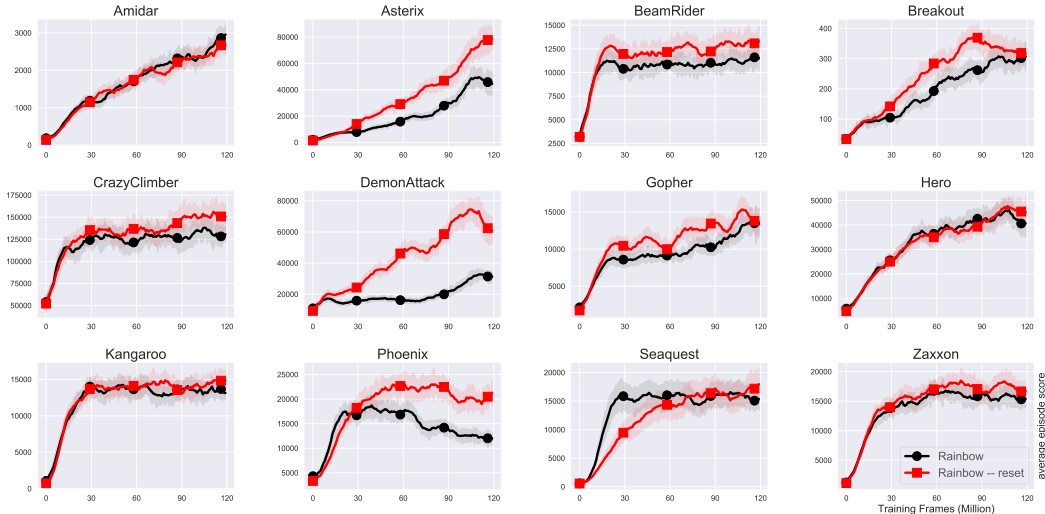

Figure 1: Performance of Rainbow with and without resetting the Adam optimizer and with the default value of $K = 8000$ on 12 randomly-chosen Atari games. All results are averaged over 10 random seeds. Resetting the optimizer often improves the agent's performance.

where this is not the case, the agent can waste many gradient updates just to "unlearn" the effects of the previous iterations on the internal parameters $m$ and $v$. This is a contamination effect that plagues RL optimization as we later demonstrate. Fortunately, there is a remarkably easy fix. Note that the Adam optimizer is fully equipped to deal with resetting the moment estimates to $\mathbf{0}$, despite the bias it introduces, because the bias is dealt with adequately by performing the debiasing step. Finally, note that this is an inexpensive and convenient fix in that it adds no computational cost to the baseline algorithm, nor does it add any new hyper-parameter.

## 4  Experiments

We chose the standard Atari benchmark [10] to perform our study, and also chose the popular Rainbow agent [5], which fruitfully combined a couple of important techniques in learning the value function. This combination resulted in Rainbow being the state-of-the-art agent, one that remains a competitive baseline to this day. We used the most popular implementation of Rainbow, namely the one in the Dopamine framework [1], and followed Dopamine's experimental protocol.

Note that other than the popular Dopamine baseline [1], we checked the second most popular implementations of DQN and Rainbow on Github, namely Github.com/devsisters/DQN-tensorflow and Github.com/Kaixhin/Rainbow, and found that resetting is absent in these implementations as well. Lack of resetting is thus not an oversight of the Dopamine implementation, but the standard practice in numerous Deep RL implementations.

### 4.1  Rainbow with Resetting Adam

Here our desire is to investigate the effect of resetting the optimizer on the behavior of the Rainbow agent with its default Adam optimizer. For all of our ablation studies, including this experiment, we worked with the following 12 games: *Amidar*, *Asterix*, *BeamRider*, *Breakout*, *CrazyClimber*, *DemonAttack*, *Gopher*, *Hero*, *Kangaroo*, *Phoenix*, *Seaquest*, and *Zaxxon*. Note that we will present comprehensive results on the full set of 55 Atari games later. Limiting our experiments to these 12 games allowed us to run multiple seeds per agent-environment pair, and therefore, obtain statistically significant results.

From Algorithm 1, note that a key hyper-parameter in the implementation of DQN (and by extension Rainbow) is the number of gradient updates ($K$) per iteration whose default value is 8000 in most deep RL papers [7, 17, 5]. We are interested to see the impact of this hyper-parameter on the performance of Rainbow with and without resetting. We first present the results for $K = 8000$ in Figure 1. We can see that with $K = 8000$ resetting the optimizer often results in improved performance.

We now change the value of $K$ to understand the impact of changing $K$ on this comparison. Lowering $K$ provides a smaller budget to the optimizer for solving each iteration. Notice that regardless of the value of $K$ we performed the same number of overall gradient updates to the optimization (online) network across the entire training. Stated differently, a smaller value of $K$ will correspond to a larger value $T$ in Algorithm 1 because for the sake of a fair comparison we always keep the multiplication of $K \times T$ fixed across all values of $K$ in Rainbow with or without resetting.

We now repeat this experiment for multiple other values of $K$. Moreover, akin to the standard practice in the literature [18], rather than looking at individual learning curves per game, hereafter we look at the human-normalized performance of the agents on all games, namely: $\frac{\text{Score}_{\text{Agent}} - \text{Score}_{\text{Random}}}{\text{Score}_{\text{Human}} - \text{Score}_{\text{Random}}}$. To compare the performance of the two agents across all 12 games, we compute the median of this number across the games and present these results for each value of $K$ in Figure 2.

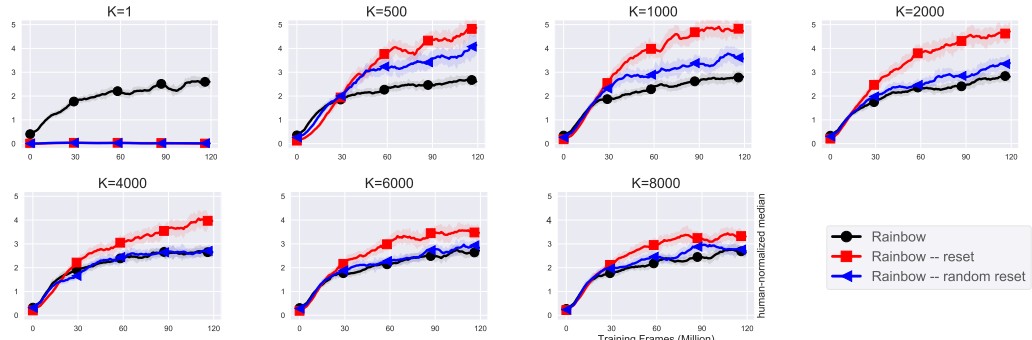

Figure 2: A comparison between Rainbow with and without resetting on the 12 Atari games for different values of $K$. The Y-axis is the human-normalized median. Observe that for all but $K = 1$ it is clearly better to reset the Adam optimizer. Notice, also that the best performance with resetting is obtained by values of $K$ that are much smaller than the default 8000 used in numerous papers.

From Figure 2 we can see very clearly that Rainbow with resetting is dominating the original Rainbow agent for all values of $K$ but $K = 1$. Notice that in the extreme case of $K = 1$ the Adam optimizer is in effect not accumulating any internal parameters because we reset after each step. Therefore, in essence, when $K = 1$ the update would be more akin to using the RProp optimizer [19], which is not effective in our setting. Another interesting trend is that in fact when we reset the optimizer, smaller values of $K$ than the default 8000 become the most competitive. This is in contrast to Rainbow without reset where the value of $K$ does not influence the final performance. This is because resetting removes the initial bias in the moment estimates that would otherwise be present when no resetting is performed. The resetting agent does not have to first "unlearn" this bias, and thus can reasonably solve each iteration with a smaller $K$ and move to the next iteration faster having accurately solved the previous iteration.

To further situate our results, we add another baseline where we reset the optimizer stochastically, meaning with some probability after each update to the optimization network. Whereas in the original resetting case (red), we reset the optimize right at the beginning of each iteration, in this case we just reset the optimizer after each update to the optimization parameter $w$, and the value of $K$ determines the probability $(1/K)$ with which we reset the Adam optimizer. With this choice of probability, on expectation we have 1 reset per iteration, but now resetting is stochastically performed at any given step. We present this result in Figure 3 where this additional baseline is indicated in blue.

We see that even randomly resetting the Adam optimizer provides some benefits relative to not resetting. However, we obtain the higher performance improvement by resetting the optimizer at the beginning of each iteration. Please see the Appendix for individual learning curves.

We further distill the above results by computing the area under the curves in Figure 2. Observe from Figure 3 that in the case of resetting (red), an inverted-U shape manifests itself with the best performance achieved by intermediate values of $K$. Also, to our surprise, in the case of no-resetting (which is indicated in black and corresponds to the standard Rainbow agent), notice that performance is basically flat as a function of $K$. Even taken to the extreme with $K = 1$, we observe no performance degradation in Rainbow. This means that freezing the target network for 8000 steps,

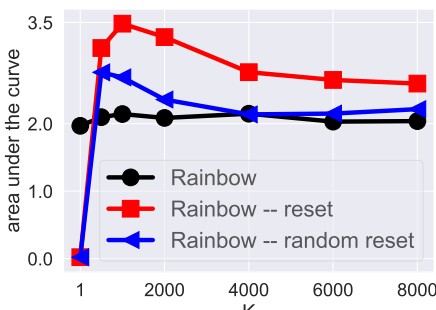

Figure 3: Effect of resetting the Adam optimizer in Rainbow. Resetting the optimizer at the beginning of each iteration yields the best performance. We also observe moderate improvements when resetting with probability $1/K$ after each update of the optimization (online) network $w$ (blue). The original Rainbow agent without resetting (black) is dominated by its resetting counterparts.

while harmless, provides no meaningful improvement to the performance of Rainbow. It is not clear to us, then, why the choice of $K = 8000$ was made in the original Rainbow paper. One possible explanation is that this choice is just a legacy from the previous papers. As we show later, in DQN it is important that a large value of $K$ is chosen, otherwise there will be extreme performance degradation, so it is possible that an assumption about the need for a large $K$ in Rainbow was made, and the validity of this assumption was not tested thoroughly.

To demonstrate the contamination effect, and to better understand where the benefit of resetting is coming from, we compute the cosine similarity between Adam's first-order moment estimate and the gradient estimate just before and after the update to the target network. Viewed as a measure of direction-wise similarity, the cosine of two vectors $u$ and $v$ is defined by $\text{Cosine}(u, v) = u^\top v/(||u|| \cdot ||v||)$. In general, $\text{Cosine}(u, v) \gg 0$ if the two vectors are similar, and $\text{Cosine}(u, v) \approx 0$ if there is no similarity. In the context of Algorithm 1, this corresponds with computing the cosine similarity between $g^i$ and $m^i$ twice per each iteration $t$, namely when $k = K - 1$ (before target update) and when $k = 0$ (after target update). We now present this result in the following table.

|  | Amidar | Asterix | BeamRider | Breakout | averaged over 12 games |
|---|---|---|---|---|---|
| before | 0.399 | 0.393 | 0.38 | 0.367 | 0.389 |
| after | -0.001 | -0.002 | -0.002 | 0.003 | -0.002 |

Table 1: Cosine similarity between Adam's first-order moment estimate and the gradient estimate just before and after updating the target network $\theta^{t+1} \leftarrow w^{t,K}$. For each game, we compute the two quantities for all iterations $t$, and then average across all $T$ iterations. We show the results for the first 4 games, and also report the average over all 12 games in the last column. Observe that the similarity is near zero after the update, meaning that direction-wise the initial moment estimate has no similarity to the gradient estimates early in the iteration. Thus, in the absence of resetting, the initial moment estimate just contaminates Adam's moment accumulation procedure.

## 4.2 Experiments with Other Agent-Optimizer Combinations

We so far demonstrated that we can improve value-function optimization in Rainbow using the idea of resetting Adam's moments. Is this effect specific to the Rainbow-Adam combination or does it generalize to settings where we use alternative optimization algorithms and even different agents?

To answer this question, we first investigate the setting where we use the RMSProp optimizer. First presented in a lecture note by Hinton [15], RMSProp can be thought of as a reduction of Adam where we only maintain a running average of the second-order moment, but not the first. In other words, RMSProp corresponds to Adam with $\beta_1 = 0$ and also skipping the debiasing step for the second-order moment estimate. Similar to Figure 3, we ran Rainbow and RMSProp with and without resetting on the same 12 games, computed the human-normalized median, and then distilled the learning curves by computing their area under the curve for each value of $K$. Our result are presented in Figure 4 (top left). Notice that while overall we see a performance degradation when we move from Adam to RMSProp, we still observe that resetting the optimizer after updating the target network can positively affect the agent's score.

We next consider a successor (rather than a predecessor) of Adam, namely the Rectified Adam optimizer introduced by Liu et al. [20]. This recent variant of Adam is especially interesting because

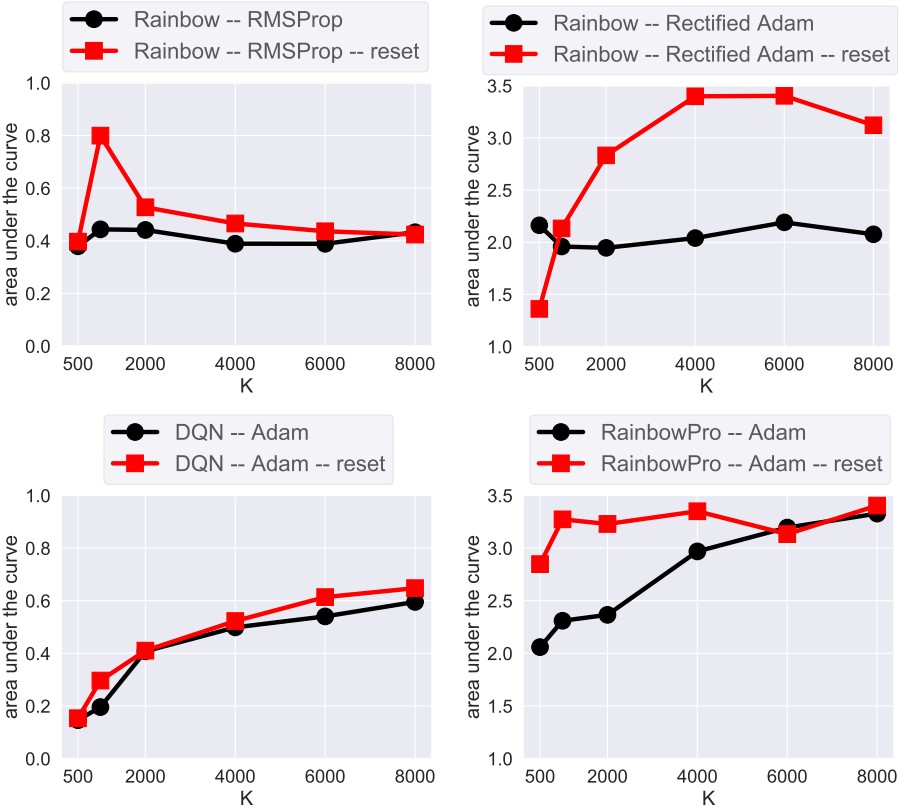

Figure 4: The positive effect of resetting in the context of Rainbow with the RMSProp optimizer (top left), Rainbow with the Rectified Adam optimizer (top right), DQN with the Adam optimizer (bottom left), and RainbowPro with the Adam optimizer (bottom right).

it controls the variance of the Adam optimizer early in training by employing step-size warm up in conjunction with a rectifying technique. With resetting, the optimization algorithm basically starts from scratch many times, and so the rectification technique of Liu et al. [20] can be crucial in further improving performance. We present this results in Figure 4 (top right). Observe that equipping the agent with resetting is having a positive impact akin to what we showed in the case of RMSprop.

Pivoting into different agents, we present the next experiment where we used the Adam optimizer in conjunction with the DQN algorithm [7]. This agent is a more primitive version of Rainbow, but we again can observe that a performance improvement manifests itself when we reset the optimizer per iteration (Figure 4 bottom left).

We next look at the RainbowPro agent introduced by Asadi et al. [21], where they endowed Rainbow with proximal updates. This ensures that at each iteration the optimization parameter gravitates towards the previous target parameter. Again, we employ resetting in the context of their introduced agent. The result, which again shows the conducive impact of resetting, is presented in Figure 4 (bottom right). Overall we can see the earlier result with Rainbow and the Adam optimizer generalizes to a couple of reasonable alternatives, thus supporting the hypothesis that our findings may remain valid for a much broader set of agent-optimizer combinations.

### 4.3 Resetting Individual Moments

So far we have shown that resetting optimizer moments positively impacts RL agents. Notice that in the specific case of Adam (and its variants), as we detailed in Section 3, two disjoint moment estimates are being computed so a natural follow up question is how important it is to reset each individual moment. Our goal is to answer this question now.

To this end, we again performed the previous experiment on the same set of 12 games. This time, we added two other resetting baselines, a first baseline in which we only reset Adam's first-order

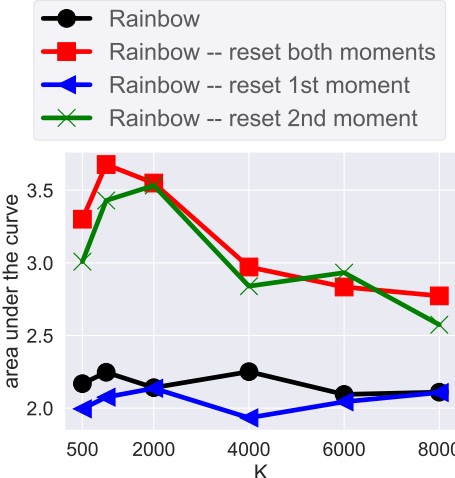

Figure 5: A comparison between different choices of resetting in Adam's moments under the Rainbow agent. The optimizer separately accumulates two moment estimates. In this context, only resetting the first-order moment (while not resetting the second-order moment) is not effective. In contrast, only resetting the second-order moment is quite effective even when we do not reset the first-order moment. The best results are obtained when resetting both the first-order and the second-order moment.

moment estimate per iteration while not resetting the second-order moment, and the converse case where we only reset the second-order moment estimate. In Figure 5, we compare the performance of these two baselines relative to the two earlier cases where we either never reset any moment, or reset both moments.

An important ingredient here is the default choice of $\beta_1$ and $\beta_2$ in popular deep-learning libraries (such as Tensorflow and PyTorch), and by extension in the Dompamine framework and in our paper. In these settings, we have $\beta_1 = 0.9$ and $\beta_2 = 0.999$. This means that the second-order moment estimate is heavily leaning on to the past gradient estimates, and so more contamination will naturally arise. This is consistent with the result here that it is quite beneficial even when we only reset the second-order moment. More generally, the importance of resetting will be somewhat dependent on the specific $\beta$ values being used. Note that one may think of our simple resetting strategy as an example of using *adaptive* $\beta$ values, where we use $\beta_1 = \beta_2 = 0$ when moving into a new iteration ($k = 0$) and then falling back into the original $\beta_1 = 0.9$ and $\beta_2 = 0.999$ for $k > 0$. Here by adaptive we mean that $\beta$ values can depend on $k$ (or even $t$) in arbitrary ways as opposed to being constant across training. We may use other adaptive strategies that can potentially improve upon our simple resetting idea, but we leave this exciting direction for future work.

### 4.4 Comprehensive Experiments on 55 Atari Games

So far we focused on performing smaller ablation studies on the subset of Atari games. We now desire to perform a more comprehensive experiment on the full 55 Atari games.

We now evaluate 3 different agent-optimizer combinations. As our benchmark, we have the original Rainbow algorithm with the Adam optimizer, without resetting, and the original value of $K = 8000$. We refer to this as the standard Rainbow agent. Our next agent is Rainbow with the Adam optimizer and resetting. In light of our results from Figure 3 we chose the value of $K = 1000$ for this agent. Finally, we have Rainbow with Rectified Adam and resetting. Similarly, and in light of Figure 4, we chose the value of $K = 4000$ for this agent. In Figure 6, we present the mean and median performance of the 3 agents on 55 games.

From Figure 6, we can see that resetting the Adam optimizer is clearly boosting the performance of the Rainbow agent in terms of both the mean and the median performance of the agent across games. Moreover, we present another comparison in terms of the per-game asymptotic improvements of the two resetting agents against the Rainbow agent in Figure 7.

### 4.5 Continuous Control

For completeness we also conducted experiments in continuous-action environments using MuJoCo physics simulator [22]. Specifically, we used Soft Actor-Critic (SAC) [23], an off-policy actor-critic algorithm with a stochastic policy. It is worth noting that SAC and Rainbow/DQN differ in several aspects. Chief among these differences is that SAC utilizes soft target updates (also known as Polyak updates) to update the target parameter, while Rainbow/DQN uses hard target updates (see Section

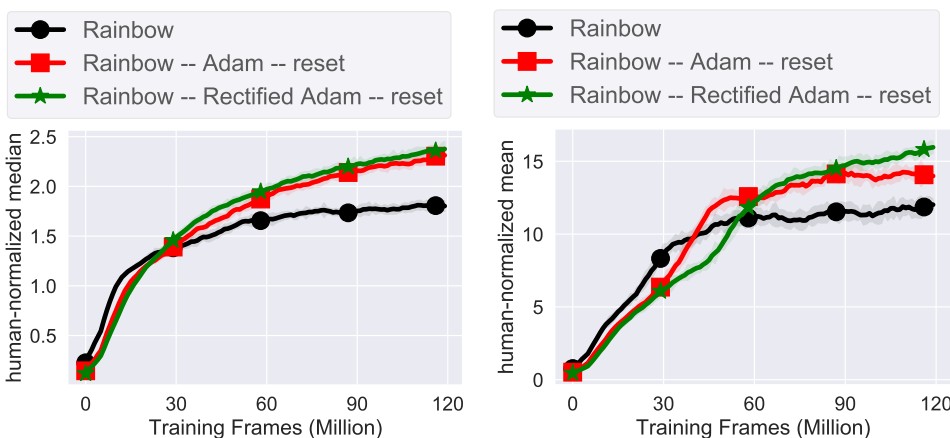

Figure 6: A comparison between Rainbow without reset against two resetting agents, namely Rainbow with restting Adam and Rainbow with resetting Rectified Adam. Results are averaged over 10 random seeds, where we human-normalize the results and take their median (left) and mean (right) over 55 games. Resetting the optimizer is clearly favorable.

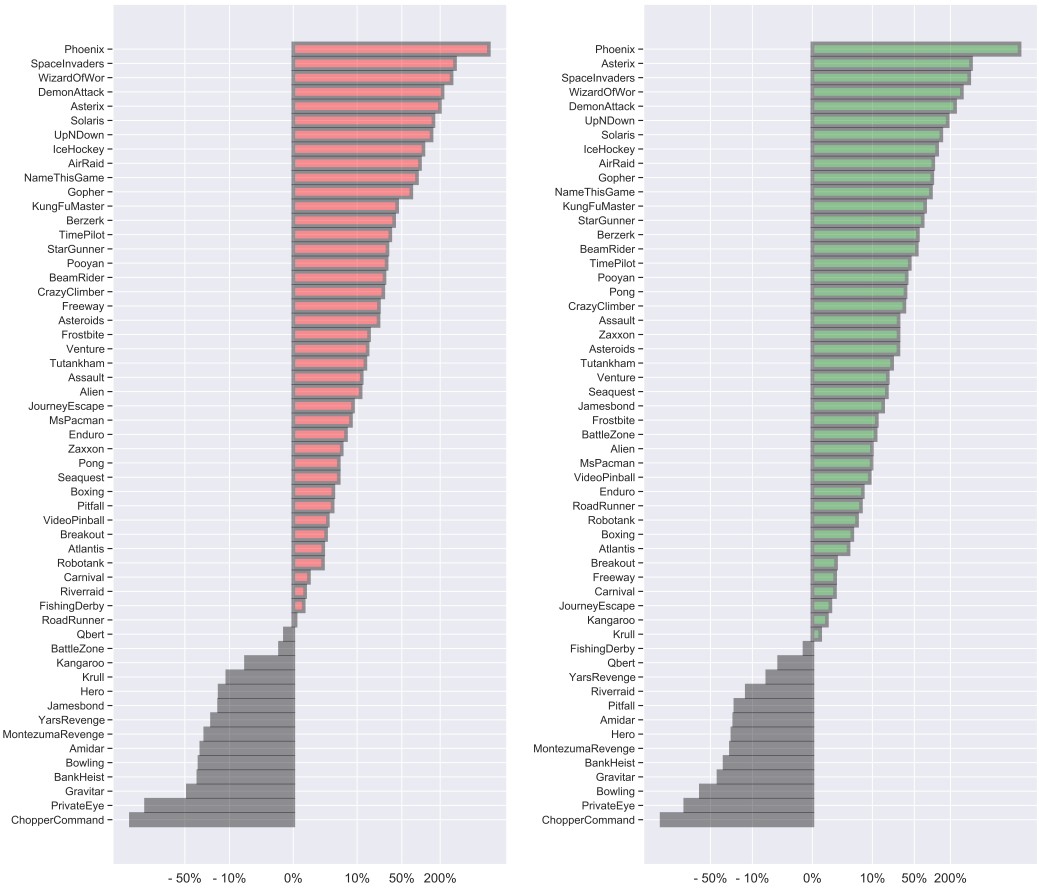

Figure 7: Asymptotic performance improvement of the resetting agents against Rainbow without resetting. In particular, we show improvements of Rainbow with resetting Adam against Rainbow (left), and Rainbow with resetting Rectified Adam against Rainbow (right).

2 for more details). SAC also uses the value of $K = 1$ so clearly it is not ideal to reset the optimizer after each step. Thus, it is not clear when to reset the optimizer. We show some our results in the Appendix, where the Adam optimizer was reset for both the critic and actor networks every 5000 steps.

It is evident that resetting the optimizer is not as helpful here to the extent it was with hard target updates, but there is still a small positive effect. Contamination is still present [4], but it is not clear how to best address it in this case, and we may require more advanced techniques than our simple resetting idea. We leave the investigation of this to future work.

## 5   Related Work

In this paper, we showed that common approaches to value-function optimization could be viewed as a sequence of optimization problems. This view is occasionally discussed in previous work [24] but in different contexts, perhaps earliest in Fitted Value Iteration [25, 26], Fitted Q Iteration [27, 28], and related approximate dynamic-programming algorithms [29, 30, 31]. More recently, Dabney et al. [32] presented the concept of the value improvement path where they argued that even in the single-task setting with stationary environments, the RL agent is still faced with a sequence of problems, and that the representation-learning process can benefit from looking at the entire sequence of problems. This view was further explored to show that ignoring this non-stationary aspect can lead to capacity [33] and plasticity loss [34]. This capacity loss problem can be mitigated in various ways, such as by regularization [33, 35, 36], introducing new parameters [34], or by using activation functions that are more conducive to learning a sequence of targets [37]. None of these works, however, considered the effect of RL non-stationarity on the internal parameters of the optimizer. Notice that this kind of non-stationarity is due to the solution not the environment, and stands in contrast to the non-stationary setting that arises when the MDP reward and transitions change, a setting that is well-explored [38, 39, 40, 41, 42, 43].

To hedge against the contamination effect, we proposed to reset the optimizer per iteration. Resetting the optimizer is a well-established idea in the optimization literature, and can be found for example in a paper by Nesterov [44]. Adaptive versions of resetting are also common and perform well in practice [45, 46]. Restarting the step-size (learning rate) also has some precedence in deep learning [47, 48, 49]. See also the standard book on large-scale optimization [50] as well as a recent review of this topic in the context of optimization [51].

Resetting RL ingredients other than the optimizer is another line of work that is related to our paper. Earliest work is due to Anderson [52], which restarts some network parameters when the magnitude of error is unusually large. A more recent example is Nikishin et al. [53] who proposed to reset some of the network weights, and the corresponding Adam statistics, to maintain plasticity. In contrast, our motivation is resetting Adam statistics is to ensure we do not contaminate the moment estimation process when we move to a new iteration. Nikishin et al. conducted their experiments in the simpler continuous-control setting with policy-gradient approaches where Polyak-based updates are often used and $K = 1$, while here we focus on resetting in the context of hard target updates with large values of $K$. Perhaps the most similar work to our paper is that of Bengio et al. [4] who noticed the harmful effect of contamination, and proposed a solution based on computing the Hessian and a Taylor expansion of the loss, one that is unfortunately too expensive to be applied in conjunction with deep networks owing to their large number of weights.

## 6   Conclusion

In this paper, we argued that value-function optimization could best be thought of as solving a sequence of optimization problems where the loss function changes per iteration. Having adopted this view, we argued that it is quite natural to reset the optimizer, which can combat the contamination effect plaguing deep RL in the absence of resetting.

A more general conclusion of our work is that we must obtain a deeper understanding of the internal process of optimization algorithms in order to unleash their true power in the context of deep RL. Standard results from optimization in supervised learning may not always readily transfer to RL. It is important to keep in mind that these optimization techniques were often designed for setting that are meaningfully different than RL, and thus some of the pitfalls of RL training can plague them when applied without proper consideration.

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

# 7 Appendix

## 7.1 Hyperparameters

We report the hyper-parameters used in our experiments.

| Rainbow-Adam hyper-parameters (shared) | |
|---|---|
| Replay buffer size | 200000 |
| Target update period | variable (default 8000) |
| Max steps per episode | 27000 |
| Batch size | 64 |
| Update period | 4 |
| Number of frame skip | 4 |
| Update horizon | 3 |
| $\epsilon$-greedy (training time) | 0.01 |
| $\epsilon$-greedy (evaluation time) | 0.001 |
| $\epsilon$-greedy decay period | 250000 |
| Burn-in period / Min replay size | 20000 |
| Discount factor ($\gamma$) | 0.99 |
| Adam learning rate | $6.25 \times 10^{-5}$ |
| Adam $\epsilon$ | 0.00015 |
| Adam $\beta_1$ | 0.9 |
| Adam $\beta_2$ | 0.999 |
| **Rainbow-RMSProp** | |
| RMSProp $\rho$ | 0.9 |
| RMSProp $\epsilon$ | $10^{-7}$ |
| **Rainbow-Rectified Adam** | |
| warm up proportion | 0.1 |
| **DQN-Adam hyper-parameters** | |
| Adam learning rate | $2 \times 10^{-4}$ |
| **RainbowPro-Adam hyper-parameters** | |
| proximal parameter ($c$) | 0.05 |

Table 2: Hyper-parameters used in our experiments.

## 7.2 Complete Results from Section 4.1

We show individual learning curves for Rainbow-Adam with and without resetting for different values of $K$.

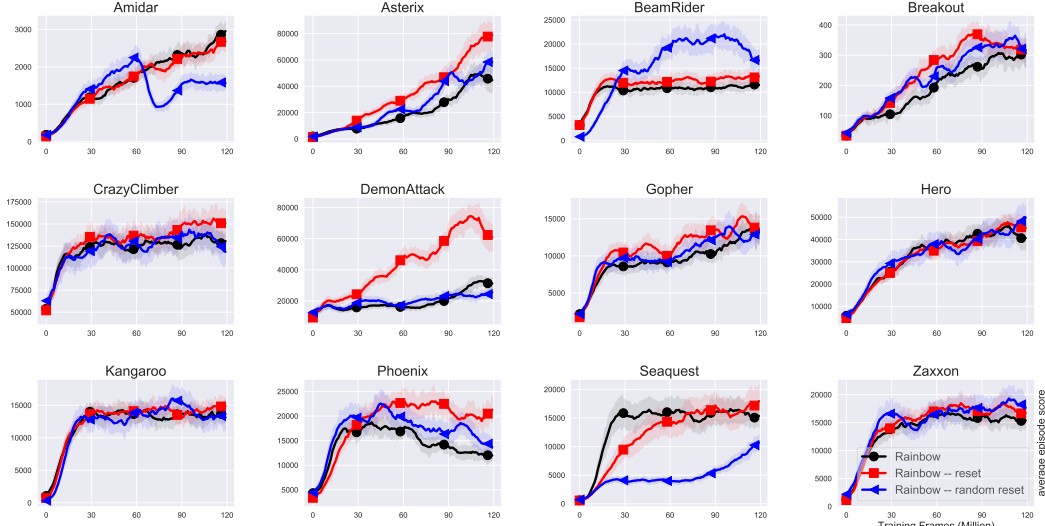

Figure 8: Learning curves for Rainbow-Adam without resetting, resetting after each target-network update, and with random resetting. In this case, $K = 8000$.

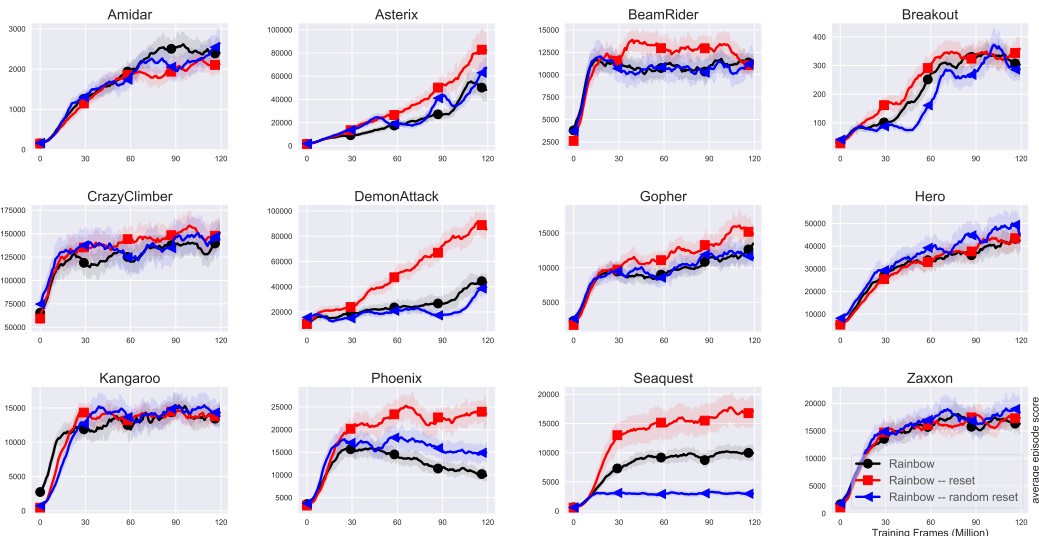

Figure 9: $K = 6000$.

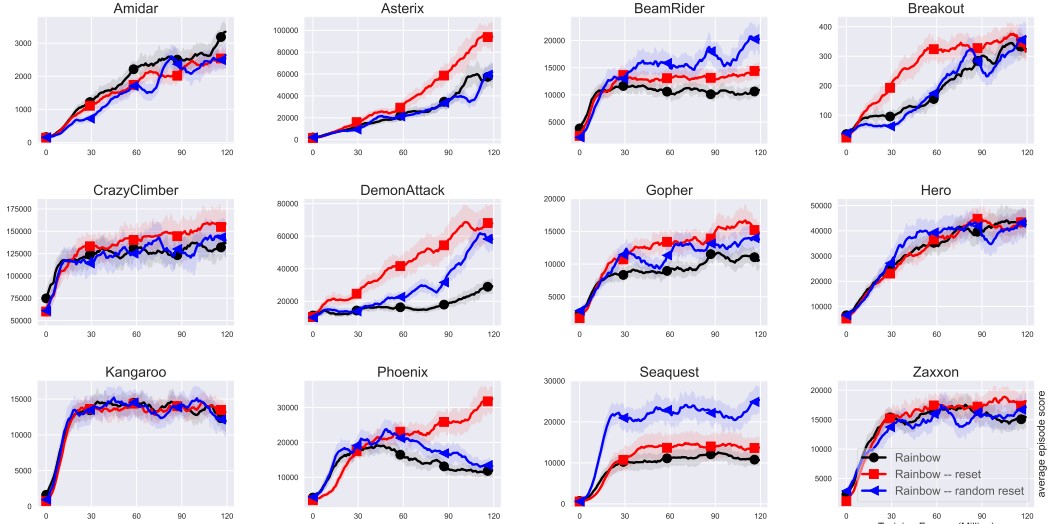

Figure 10: $K = 4000$.

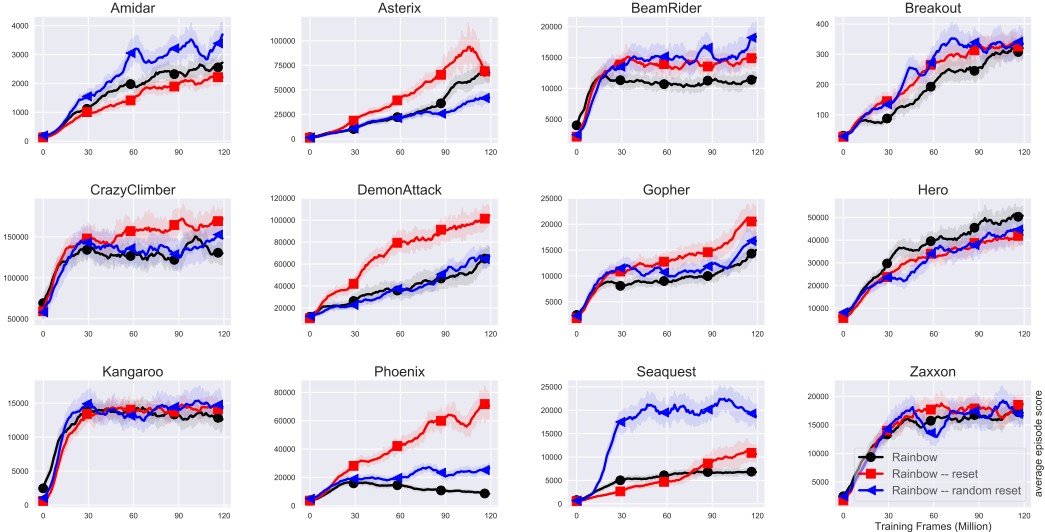

Figure 11: $K = 2000$.

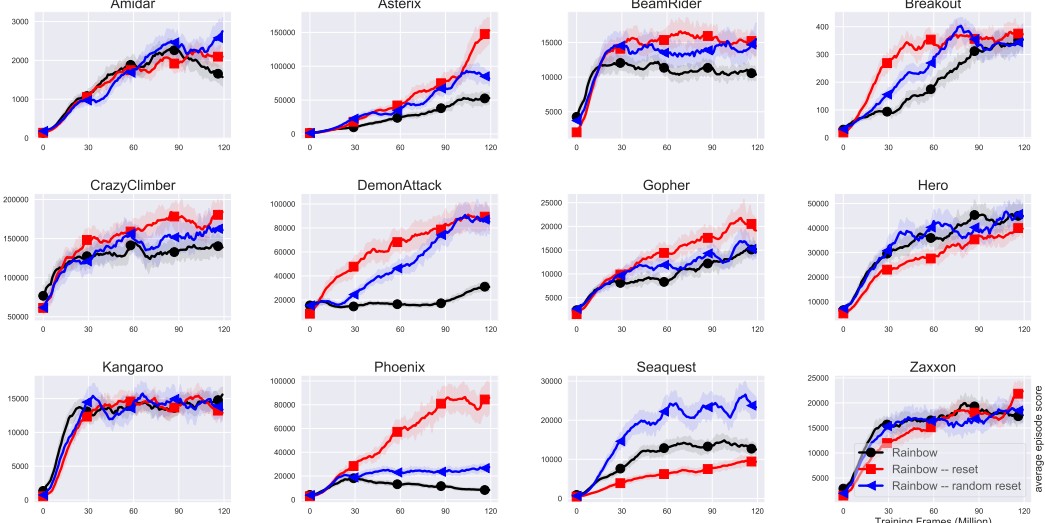

Figure 12: $K = 1000$.

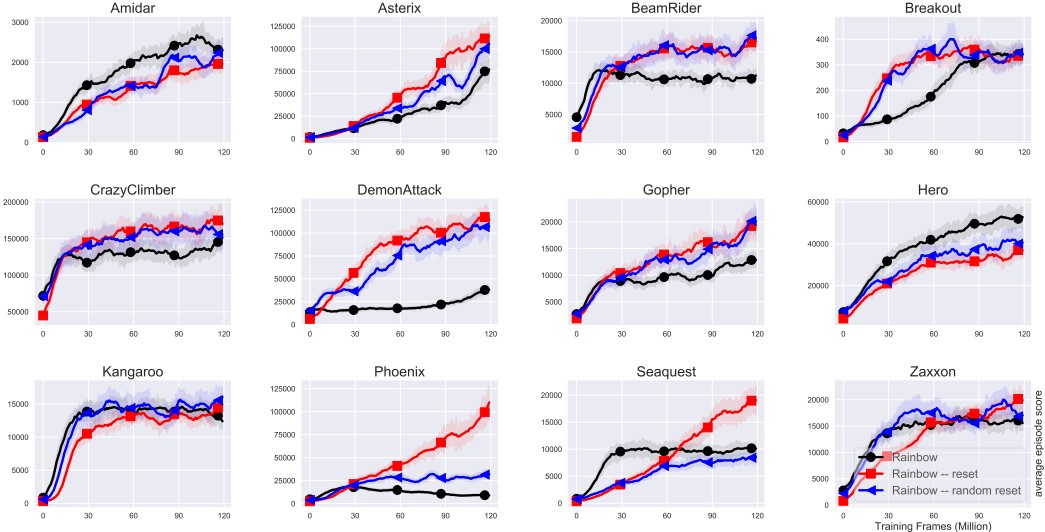

Figure 13: $K = 500$.

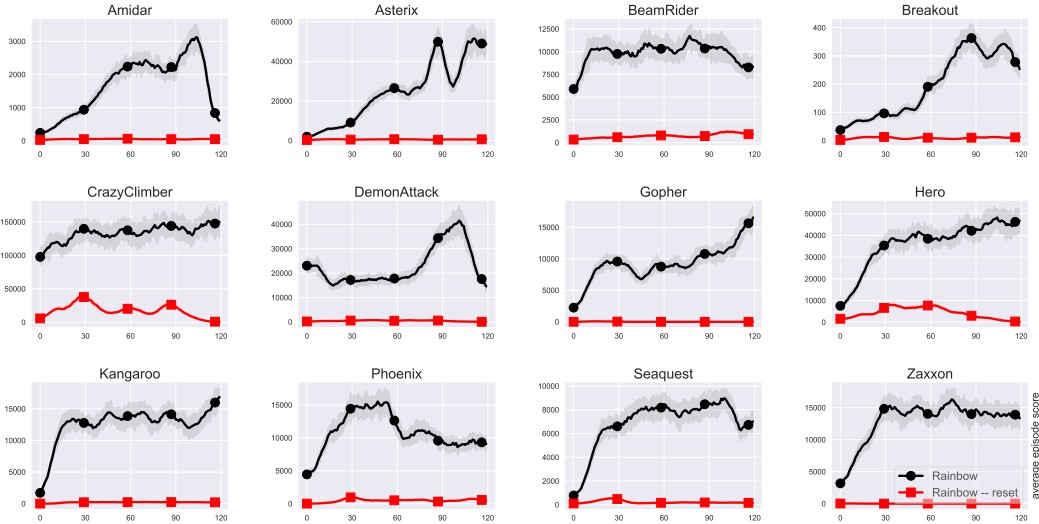

Figure 14: $K = 1$. With this value of $K$, two of the agents (Rainbow – reset and Rainbow – random reset) become identical.

## 7.3 Complete Results from Section 4.2

We now show complete results from Section 4.2 starting with Rainbow RMSProp.

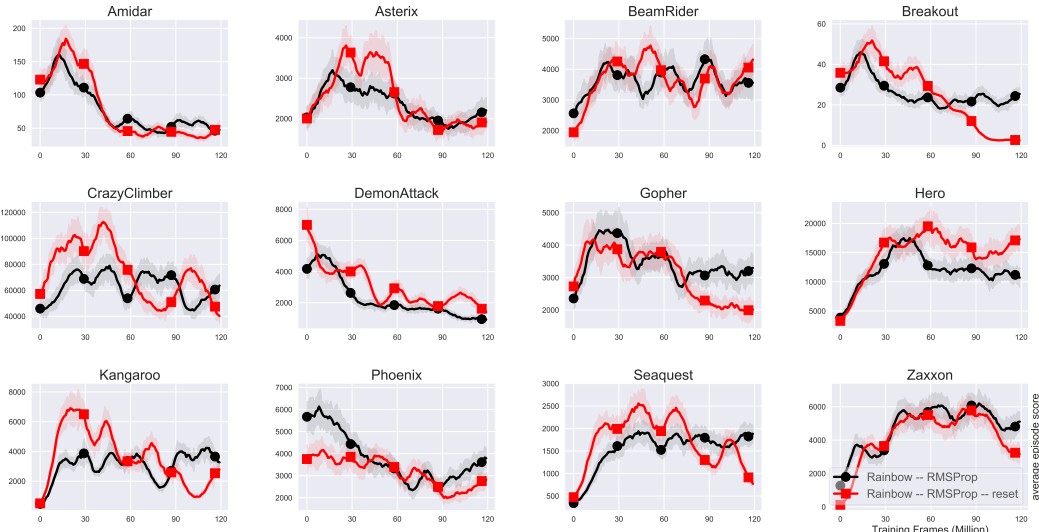

Figure 15: Performance of Rainbow with and without resetting the RMSProp optimizer and with a fixed value of $K = 8000$ on 12 randomly-chosen Atari games.

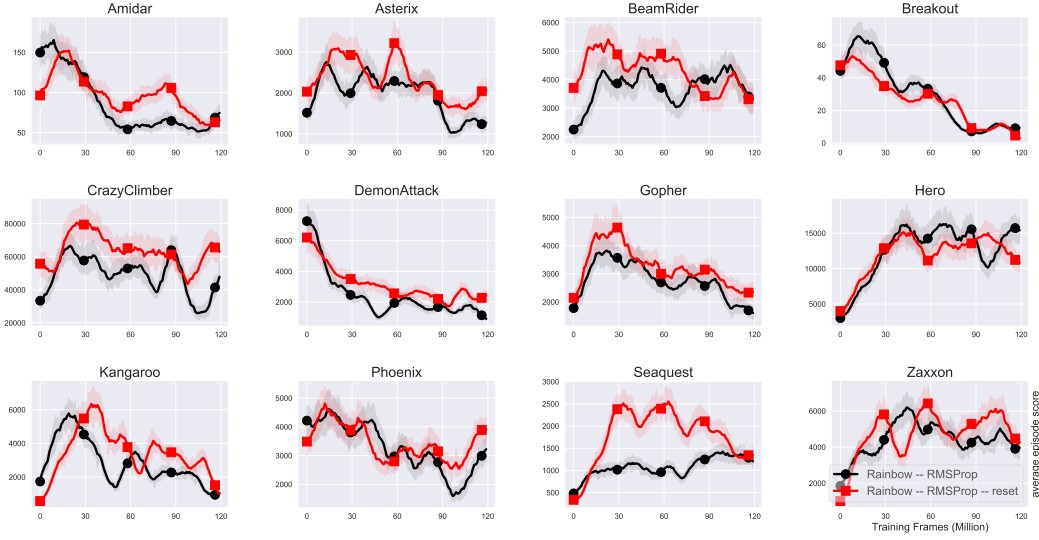

Figure 16: $K = 6000$.

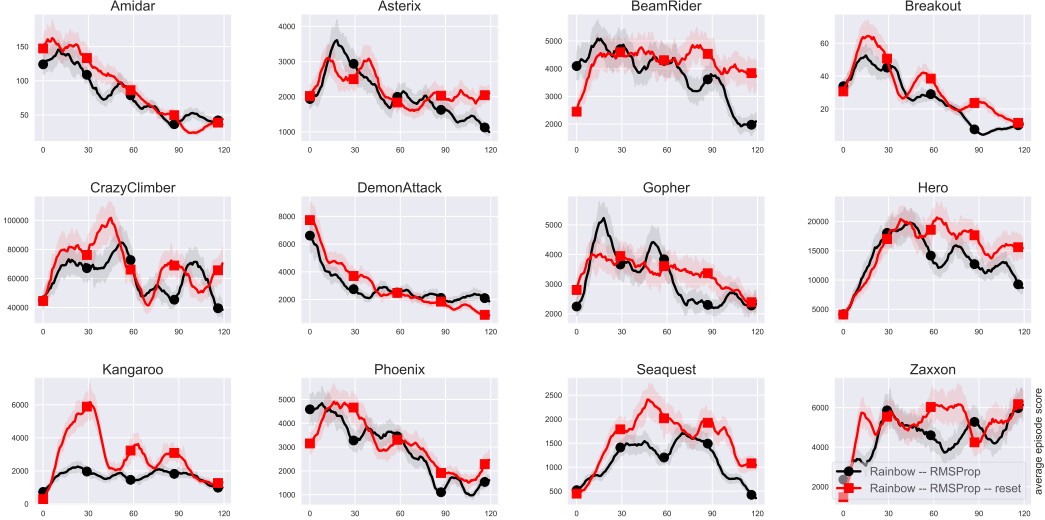

Figure 17: $K = 4000$.

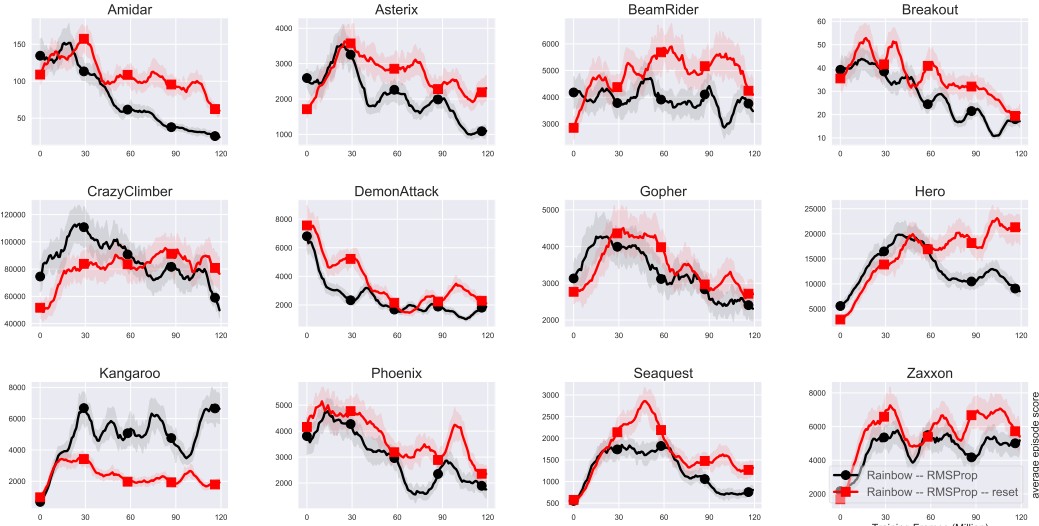

Figure 18: $K = 2000$.

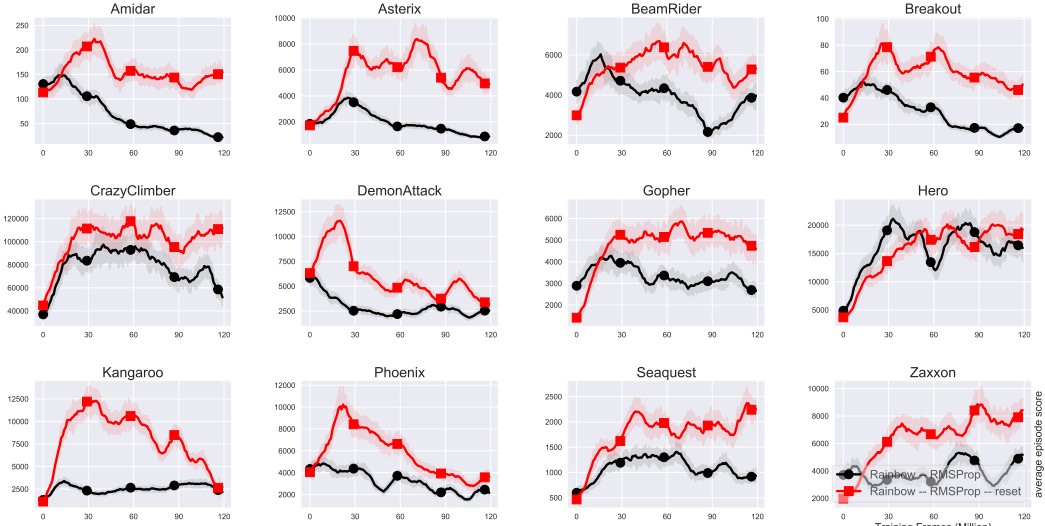

Figure 19: $K = 1000$.

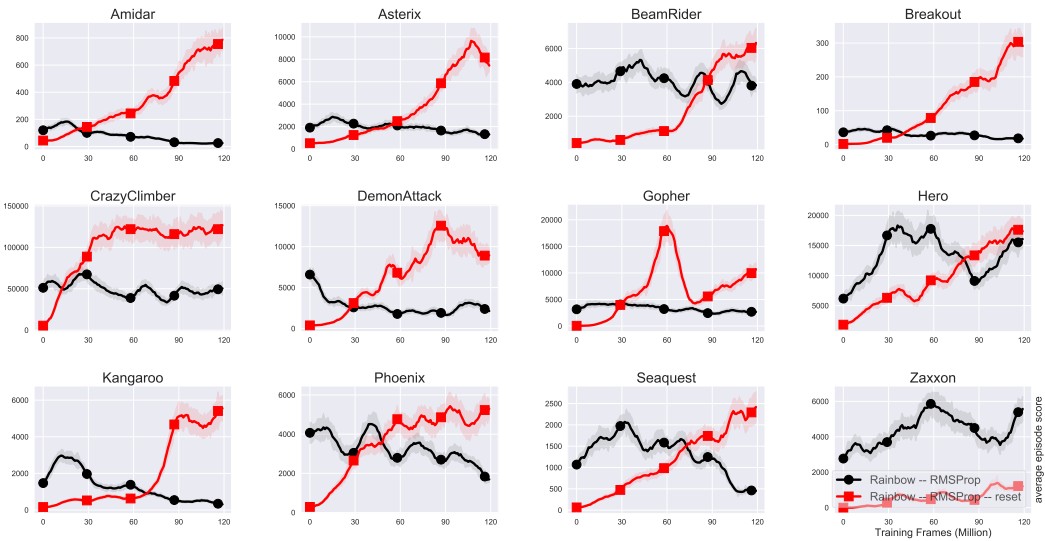

Figure 20: $K = 500$.

We now take the human-normalized median on 12 games and present them for each value of $K$.

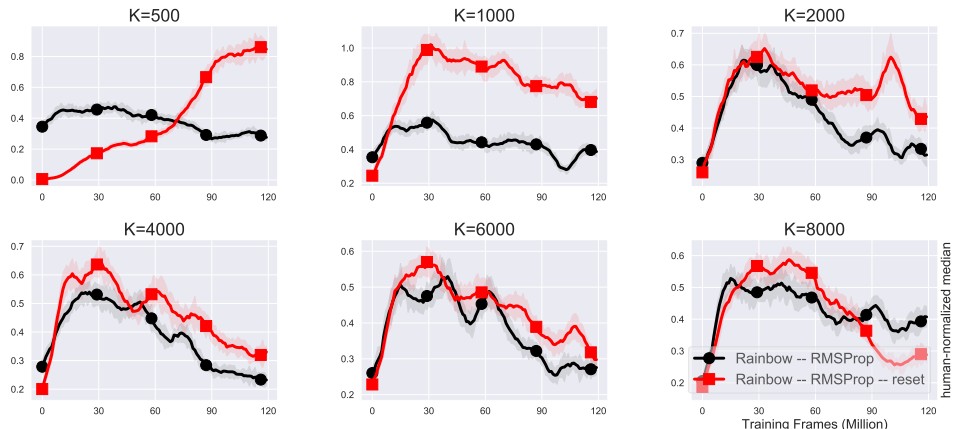

Figure 21: A comparison between Rainbow with and without resetting RMSProp on the 12 Atari games for different values of $K$.

We now look at Rainbow with the Rectified Adam optimizer.

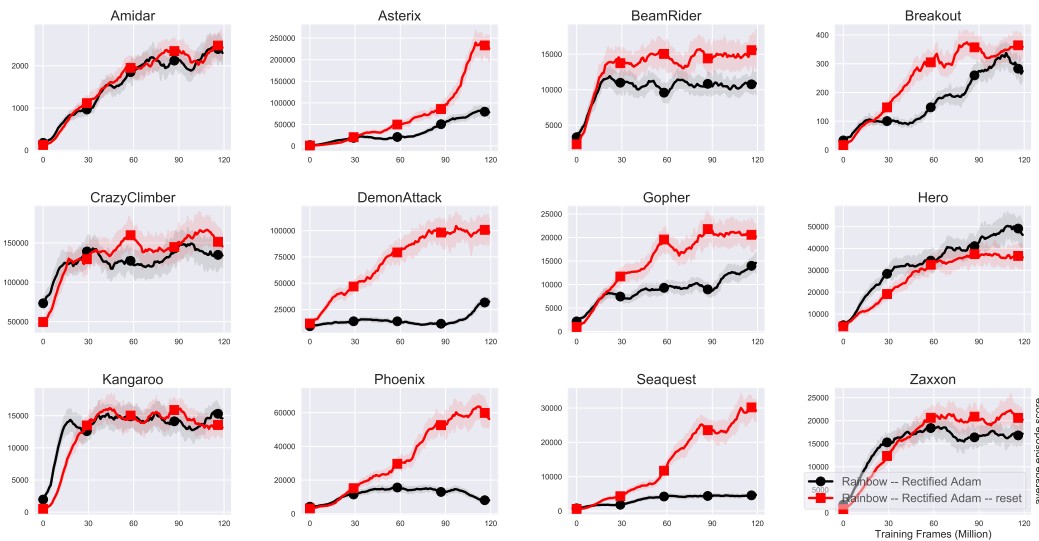

Figure 22: Performance of Rainbow with and without resetting the Rectified Adam optimizer and with a fixed value of $K = 8000$ on 12 randomly-chosen Atari games.

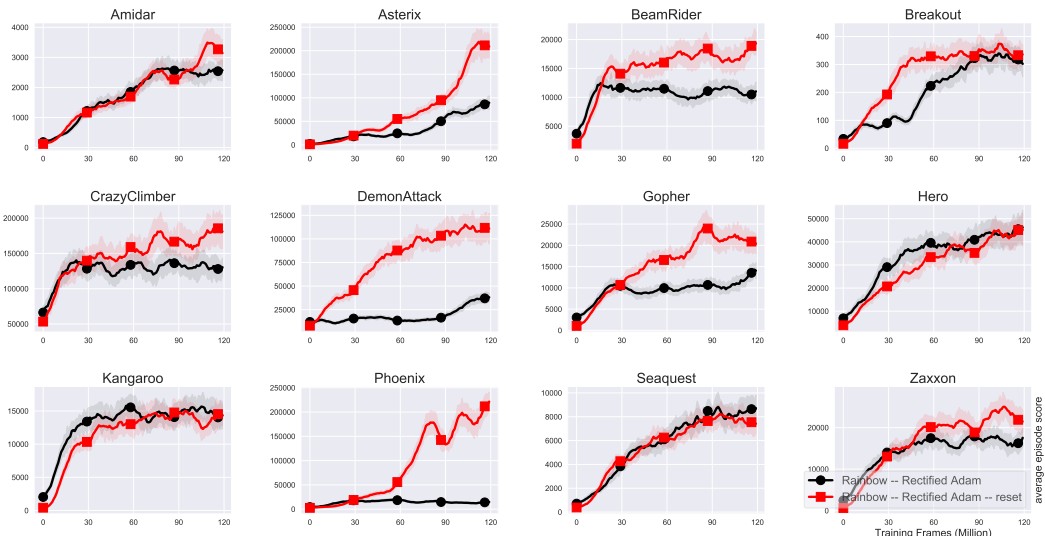

Figure 23: $K = 6000$.

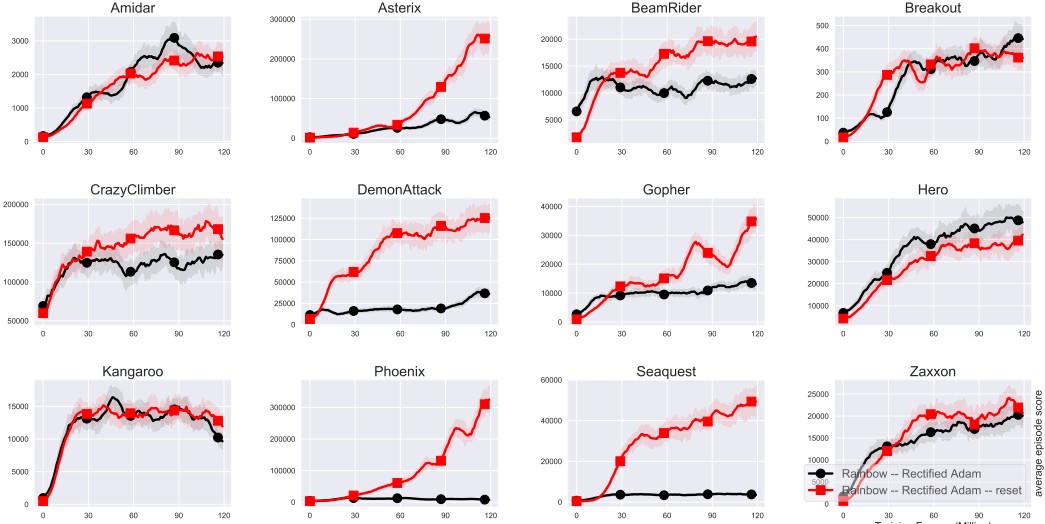

Figure 24: $K = 4000$.

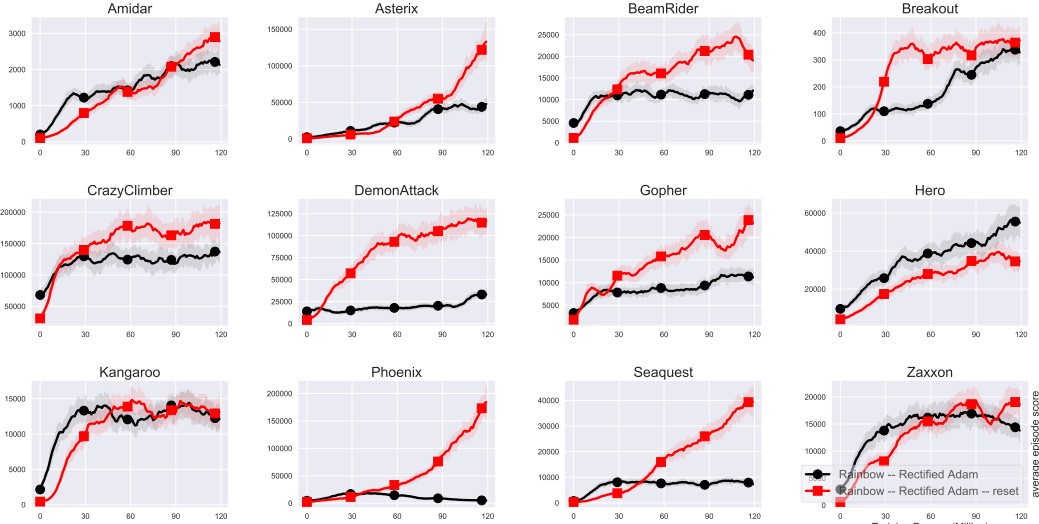

Figure 25: $K = 2000$.

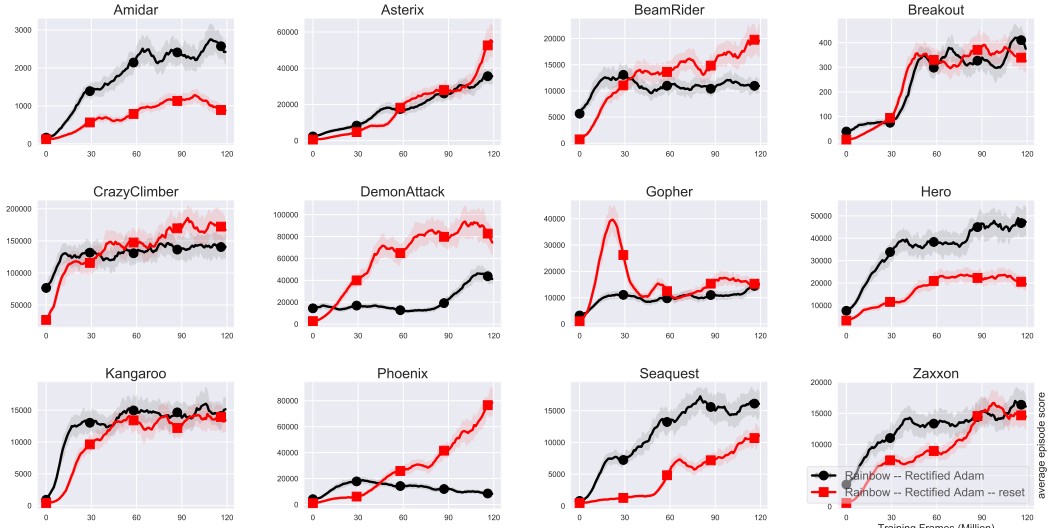

Figure 26: $K = 1000$.

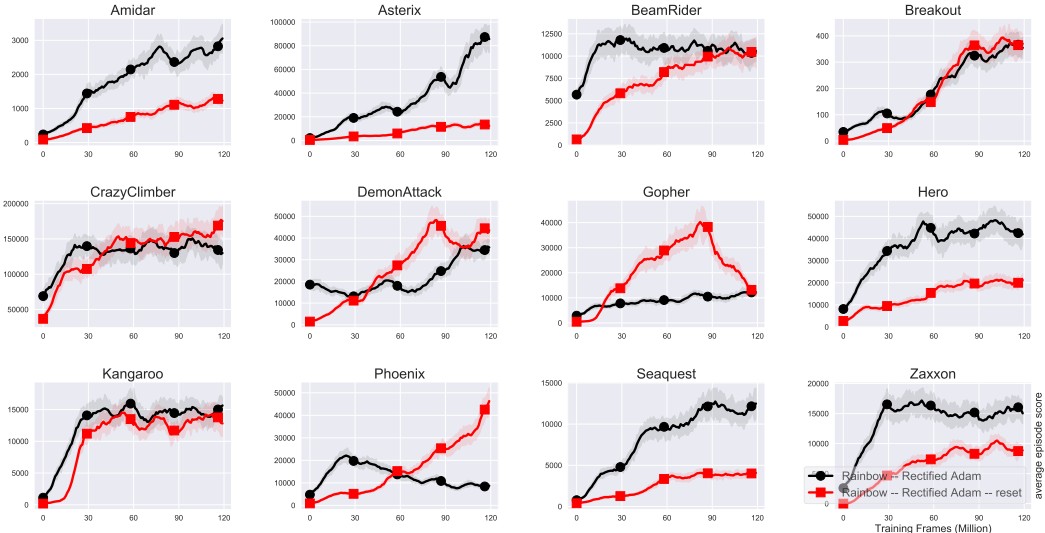

Figure 27: $K = 500$.

We now take the human-normalized median on 12 games and present them for each value of $K$.

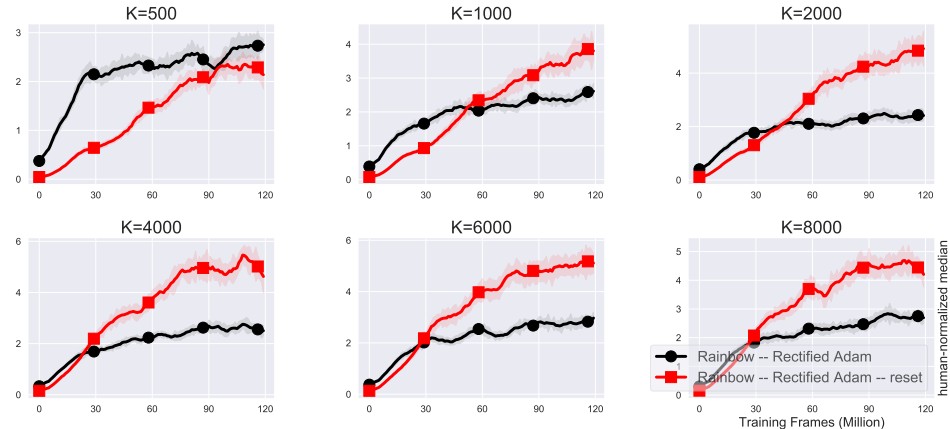

Figure 28: A comparison between Rainbow with and without resetting Rectified Adam on the 12 Atari games for different values of $K$.

We now move to the case of DQN-Adam.

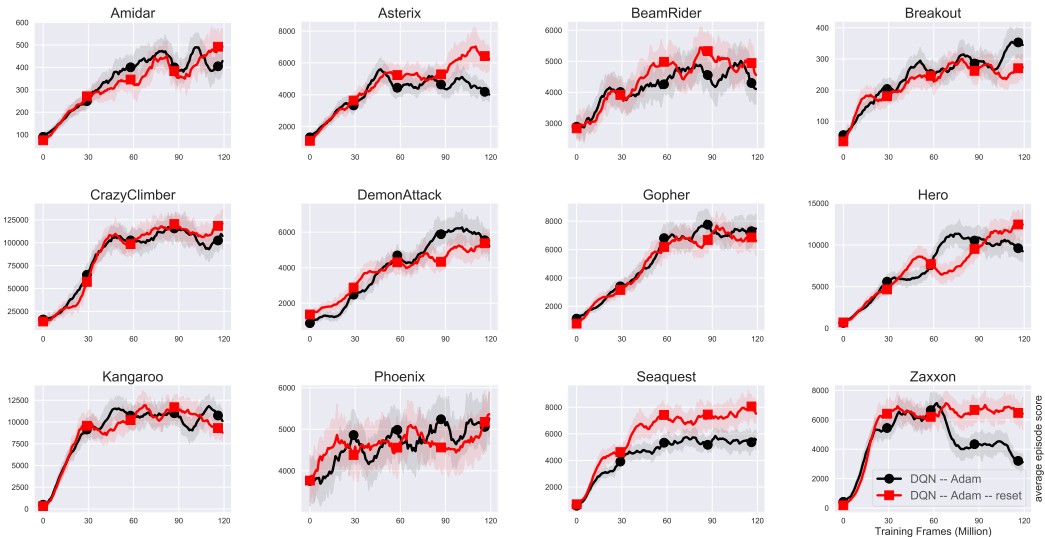

Figure 29: Performance of DQN with and without resetting the Adam optimizer and with a fixed value of $K = 8000$ on 12 randomly-chosen Atari games.

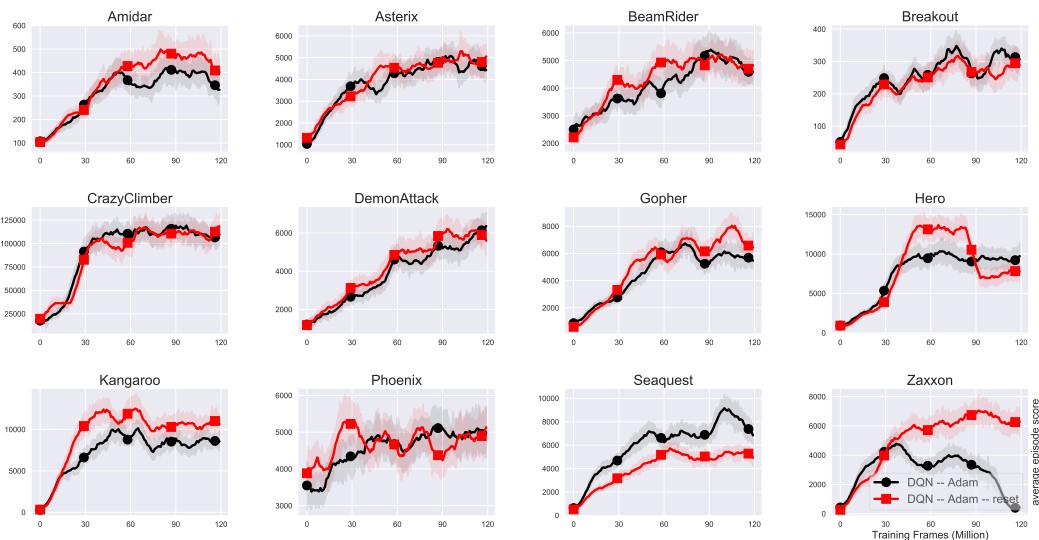

Figure 30: $K = 6000$.

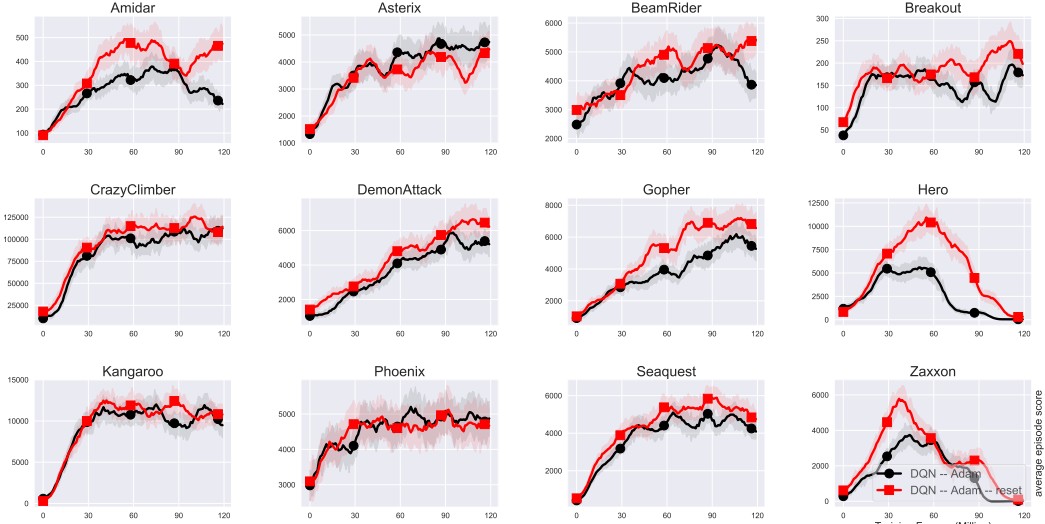

Figure 31: $K = 4000$.

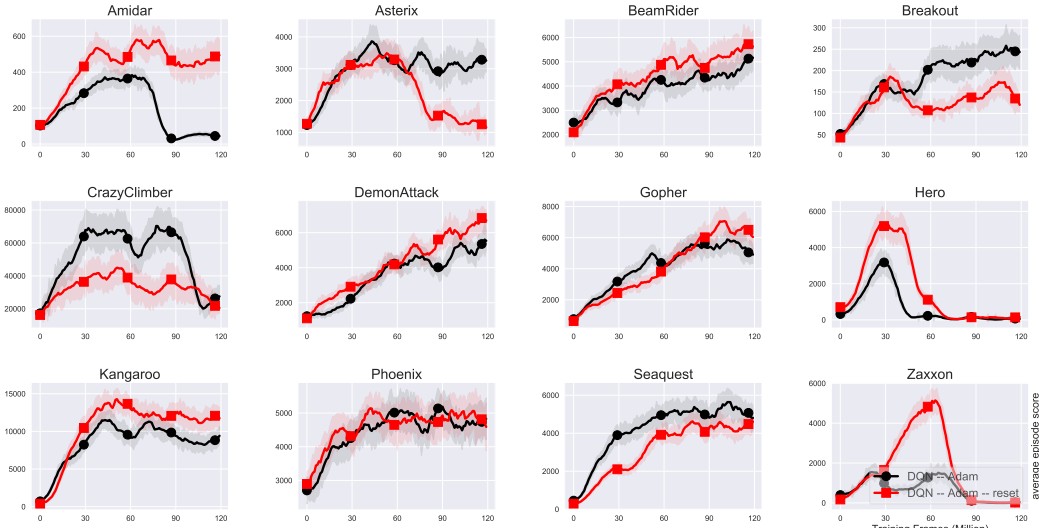

Figure 32: $K = 2000$.

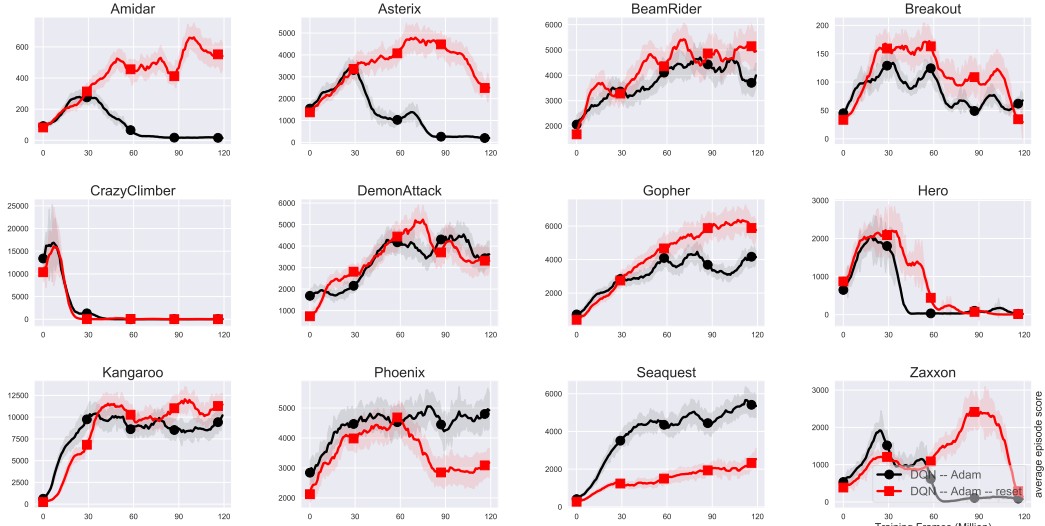

Figure 33: $K = 1000$.

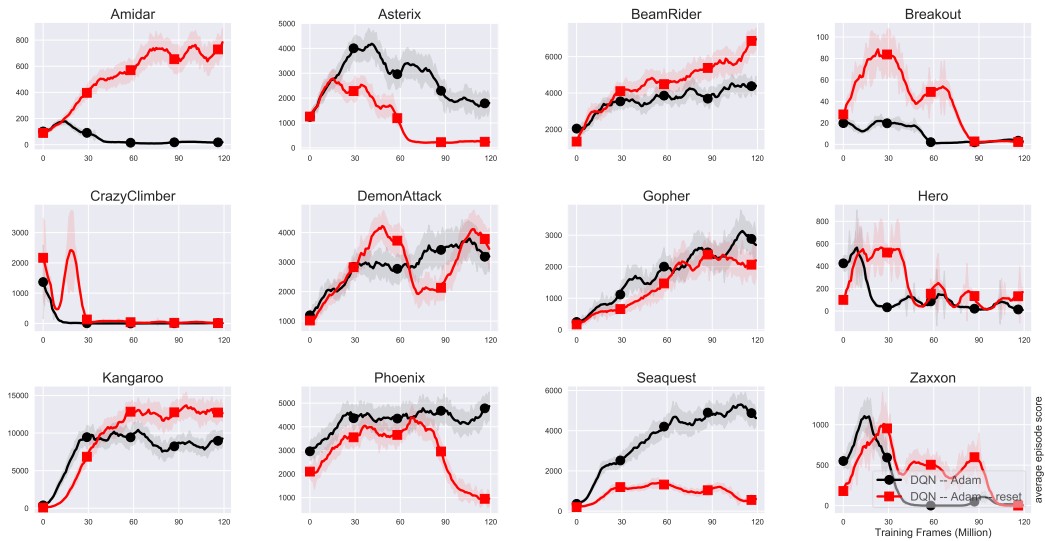

Figure 34: $K = 500$.

We now take the human-normalized median on 12 games and present them for each value of $K$.

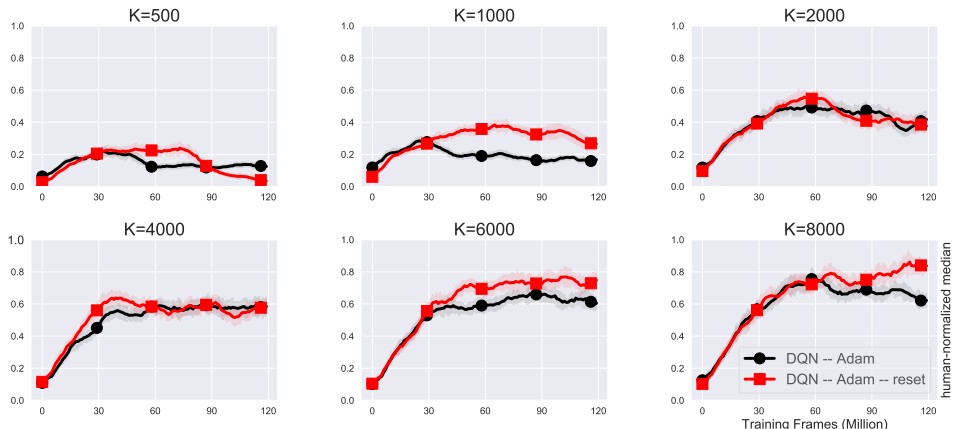

Figure 35: A comparison between DQN with and without resetting Adam on the 12 Atari games for different values of $K$.

We now move to the Rainbow Pro agent with Adam.

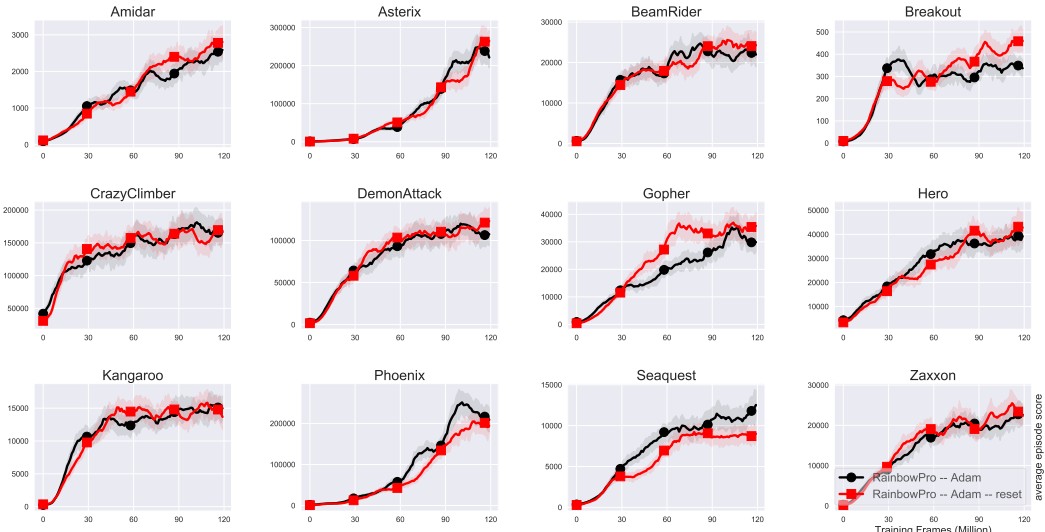

Figure 36: Performance of Rainbow Pro with and without resetting the Adam optimizer and with a fixed value of $K = 8000$ on 12 randomly-chosen Atari games.

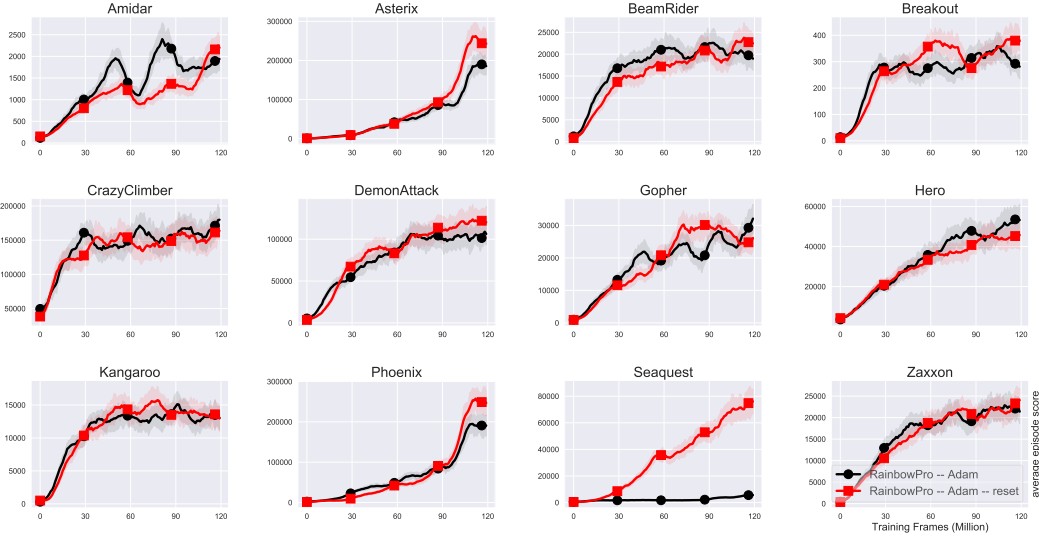

Figure 37: $K = 6000$.

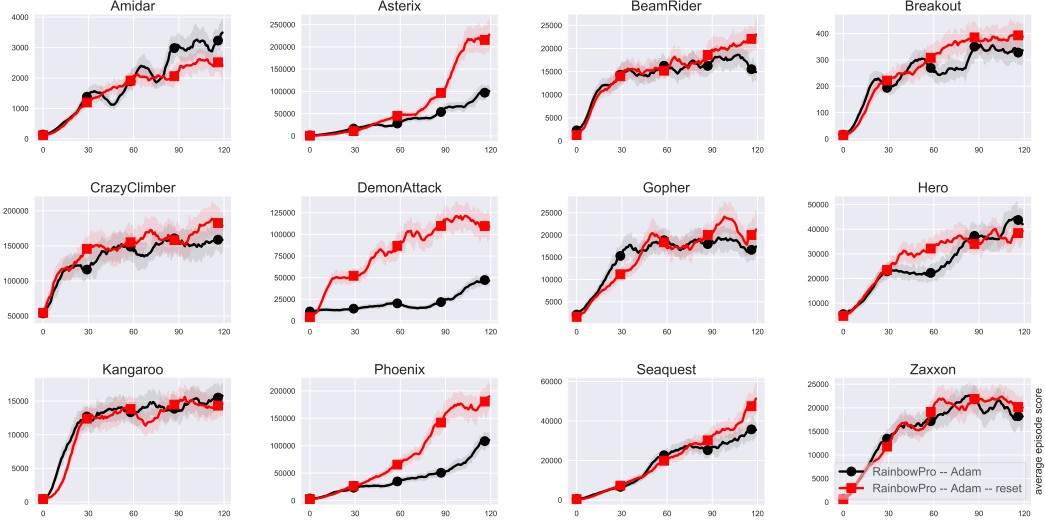

Figure 38: $K = 4000$.

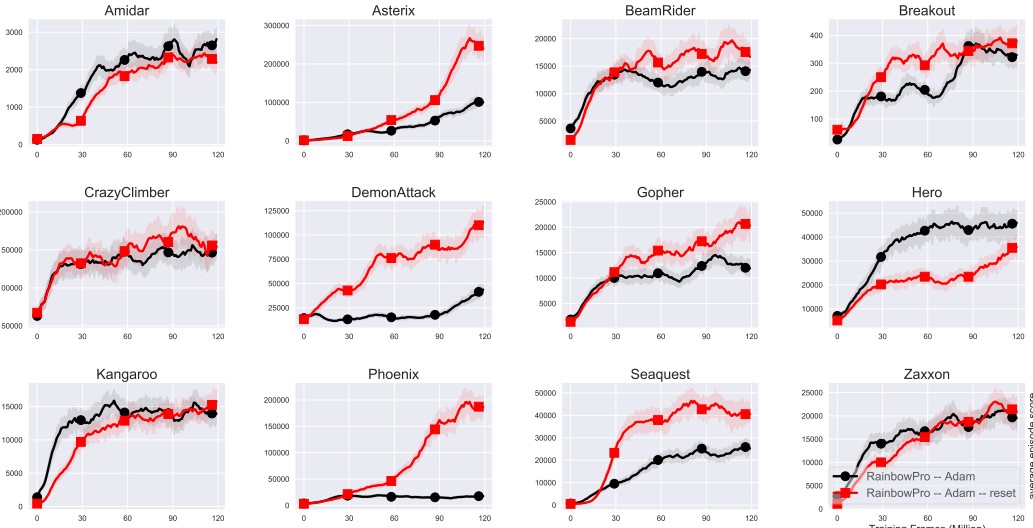

Figure 39: $K = 2000$.

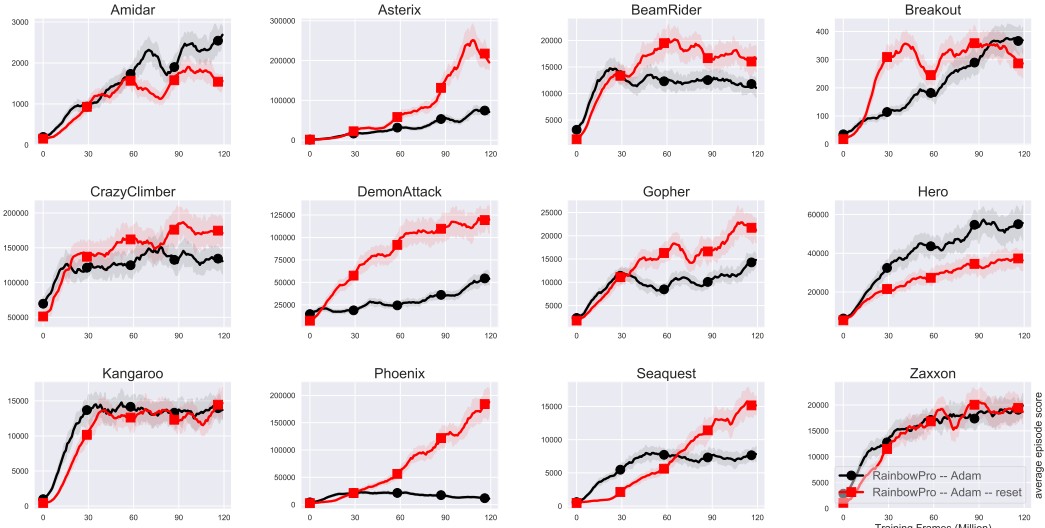

Figure 40: $K = 1000$.

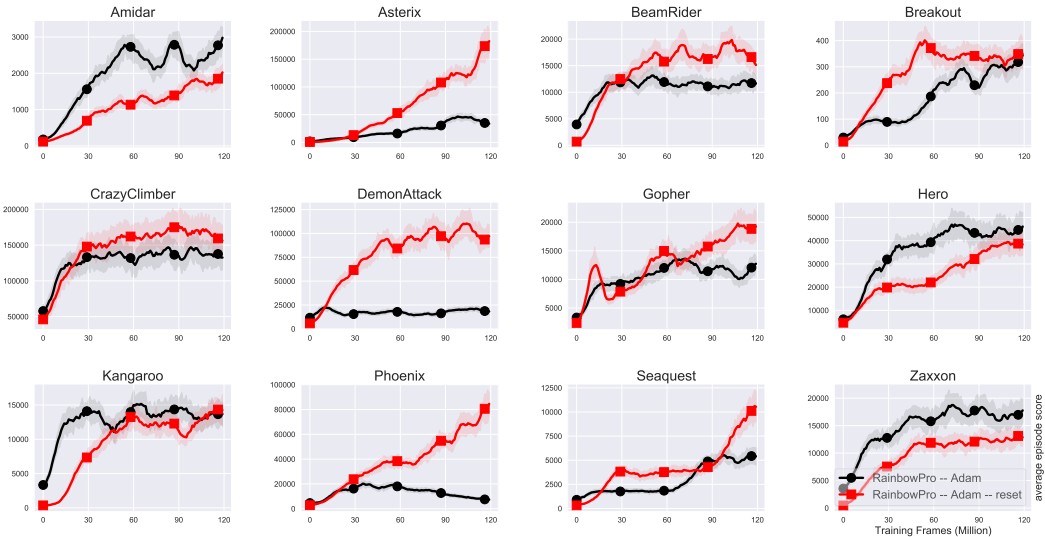

Figure 41: $K = 500$.

We now take the human-normalized median on 12 games and present them for each value of $K$.

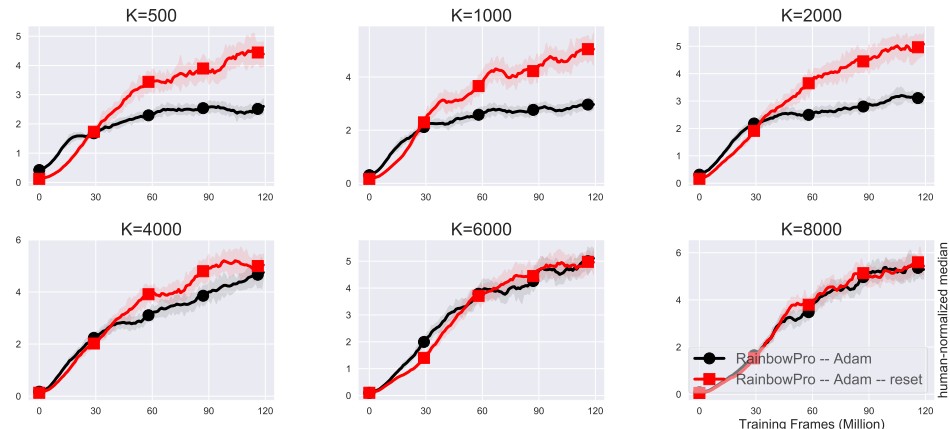

Figure 42: A comparison between Rainbow Pro with and without resetting Adam on the 12 Atari games for different values of $K$.

## 7.4 Complete Results from Section 4.3

We now show the full learning curves pertaining to Section 4.3 where we studied resetting individual moments.

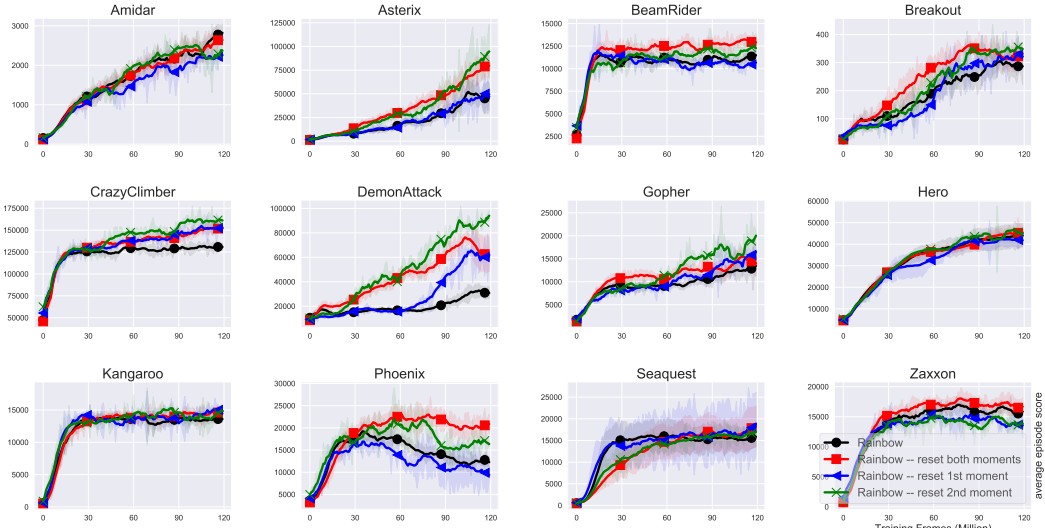

Figure 43: Learning curves for Rainbow-Adam without resetting, resetting after each target-network update, and with random resetting. In this case, $K = 8000$.

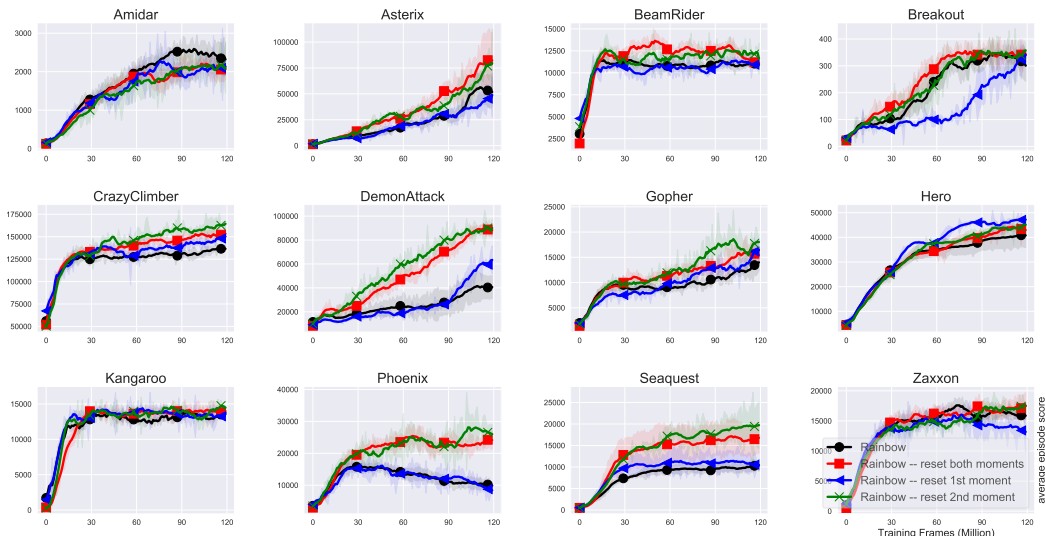

Figure 44: $K = 6000$.

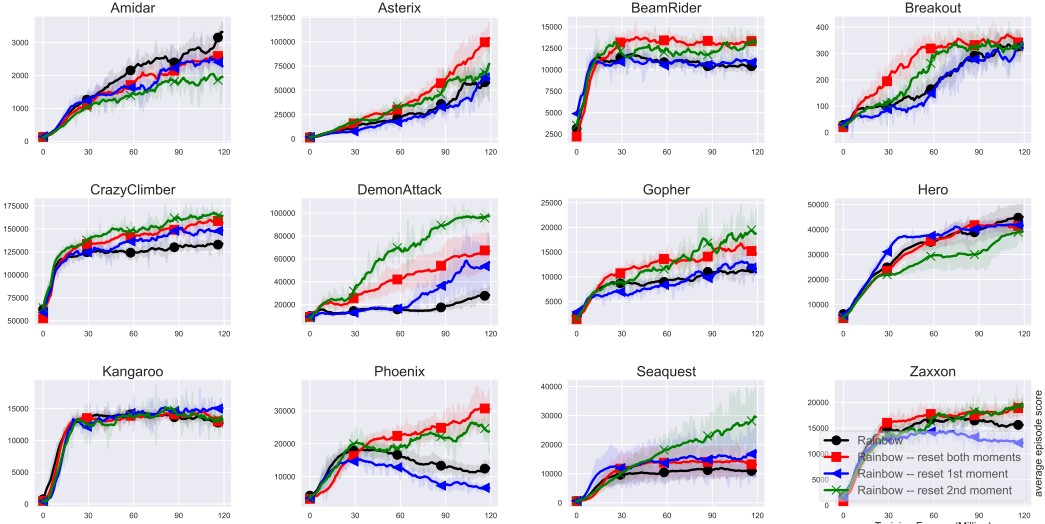

Figure 45: $K = 4000$.

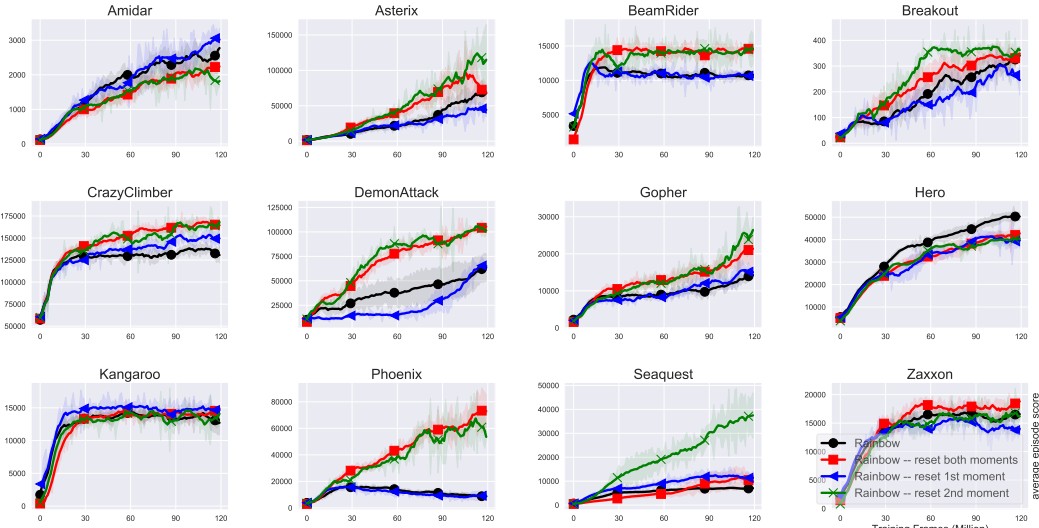

Figure 46: $K = 2000$.

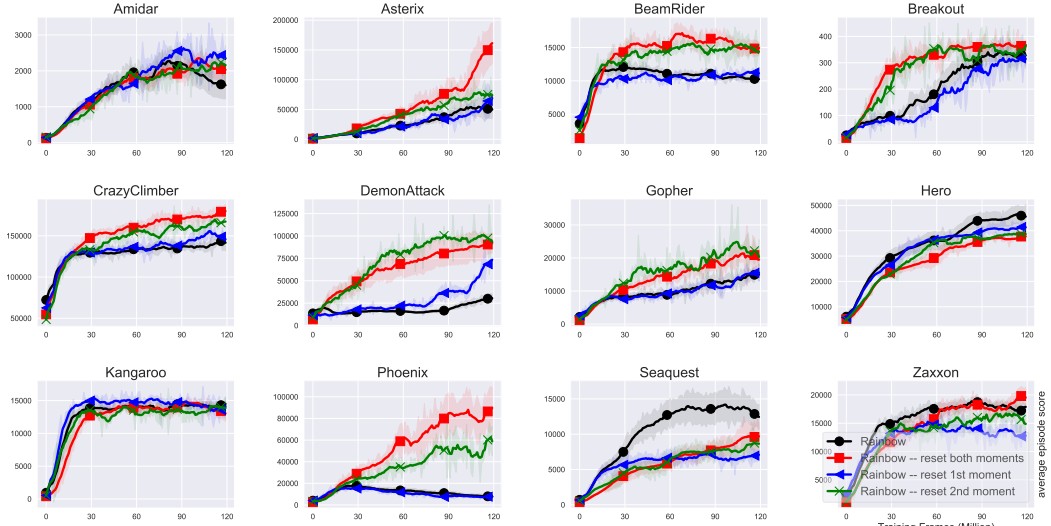

Figure 47: $K = 1000$.

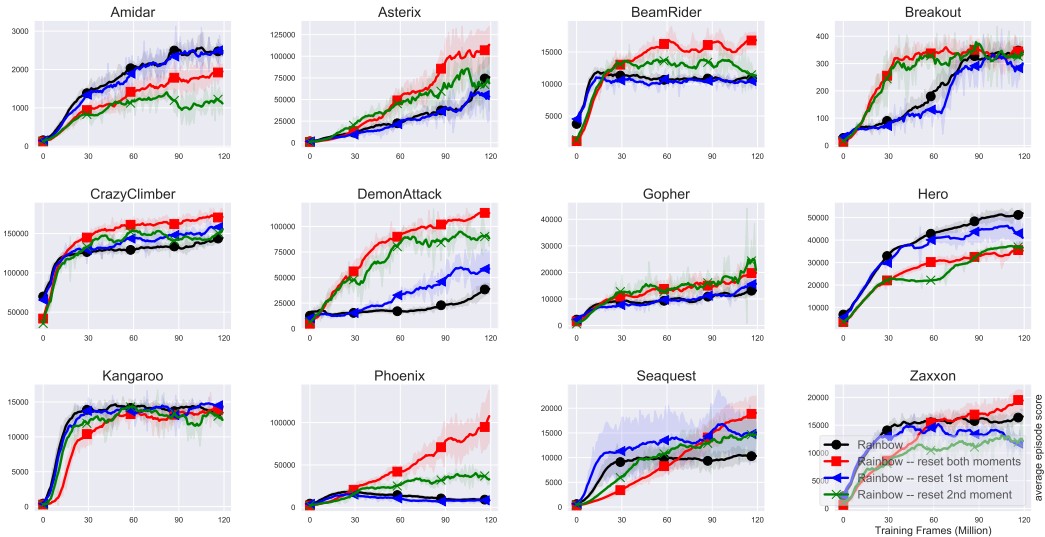

Figure 48: $K = 500$.

We now take the human-normalized median on 12 games and present them for each value of $K$.

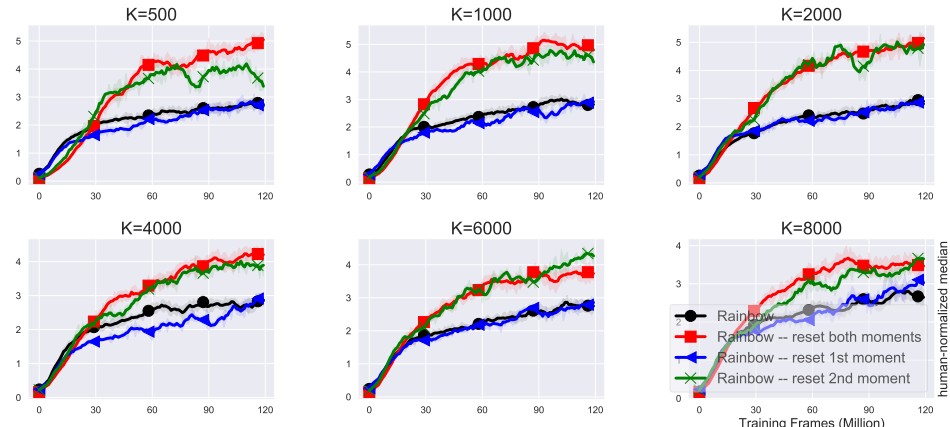

Figure 49: A comparison between no resetting, resetting both moments, and resetting individual moments in Rainbow-Adam.

## 7.5 Complete Results From Section 4.4

We now show full learning curves for all 55 Atari games and over 10 random seeds. We benchmark three agents: the default Rainbow agent from the Dopamine (no reset), Rainbow with resetting the Adam optimizer, and Rainbow with resetting the rectified Adam optimizer.

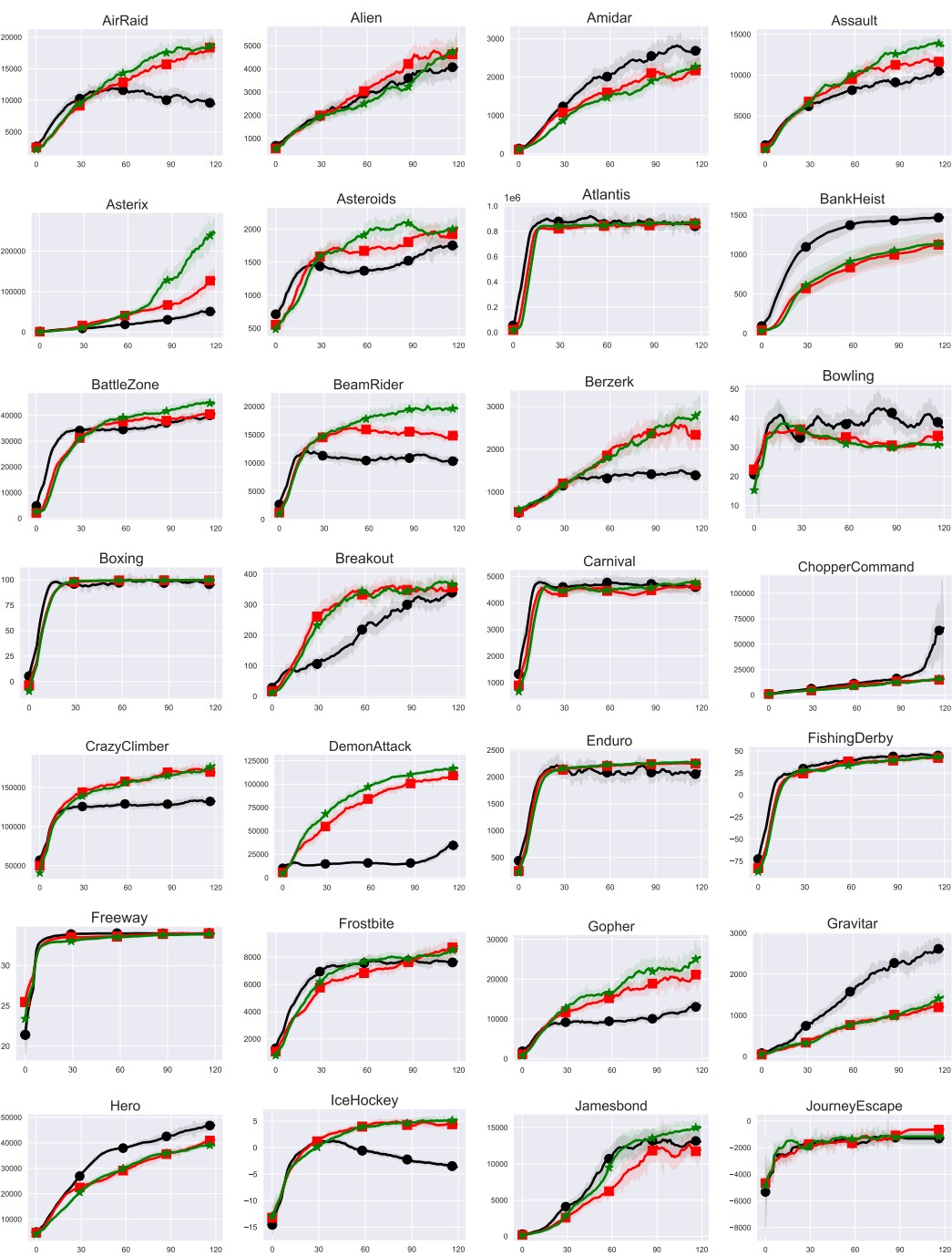

Figure 50: Learning curves for 55 games (Part I).

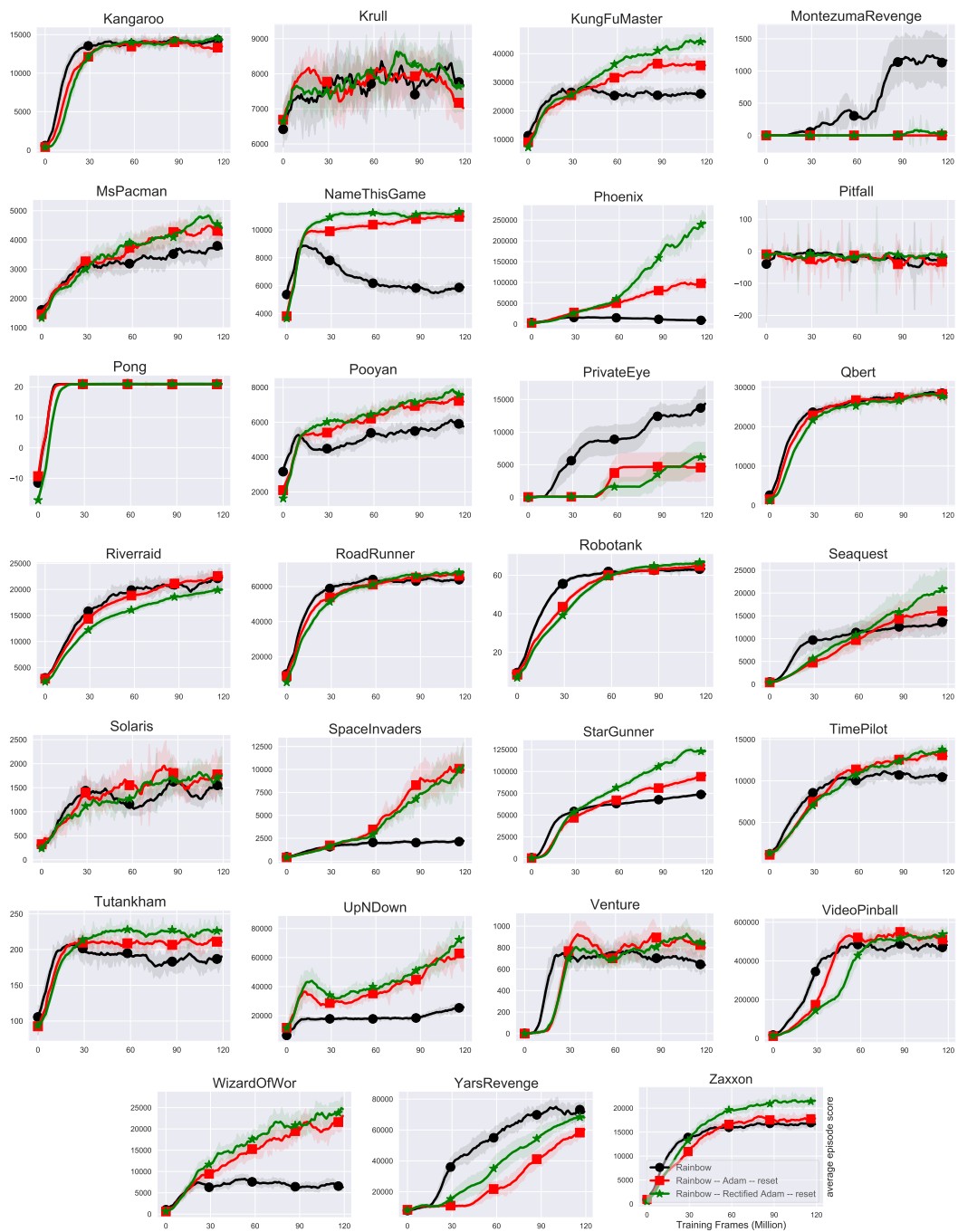

Figure 51: Learning curves for 55 games (Part II).

## 7.6 Complete Results From Section 4.5

We finally present results on continuous control task with soft actor critic (SAC) and the Adam optimizer, where we reset the optimizers every 5000 steps. Note that, in contrsat to Atari and Rainbow, the target parameter $\theta$ is updated using the Polyak strategy, so it is less clear when to reset the optimizer. Thus we chose the simple strategy of resetting the optimizer every 5000 steps. We leave further exploration of resetting with Polyak updates to future work.

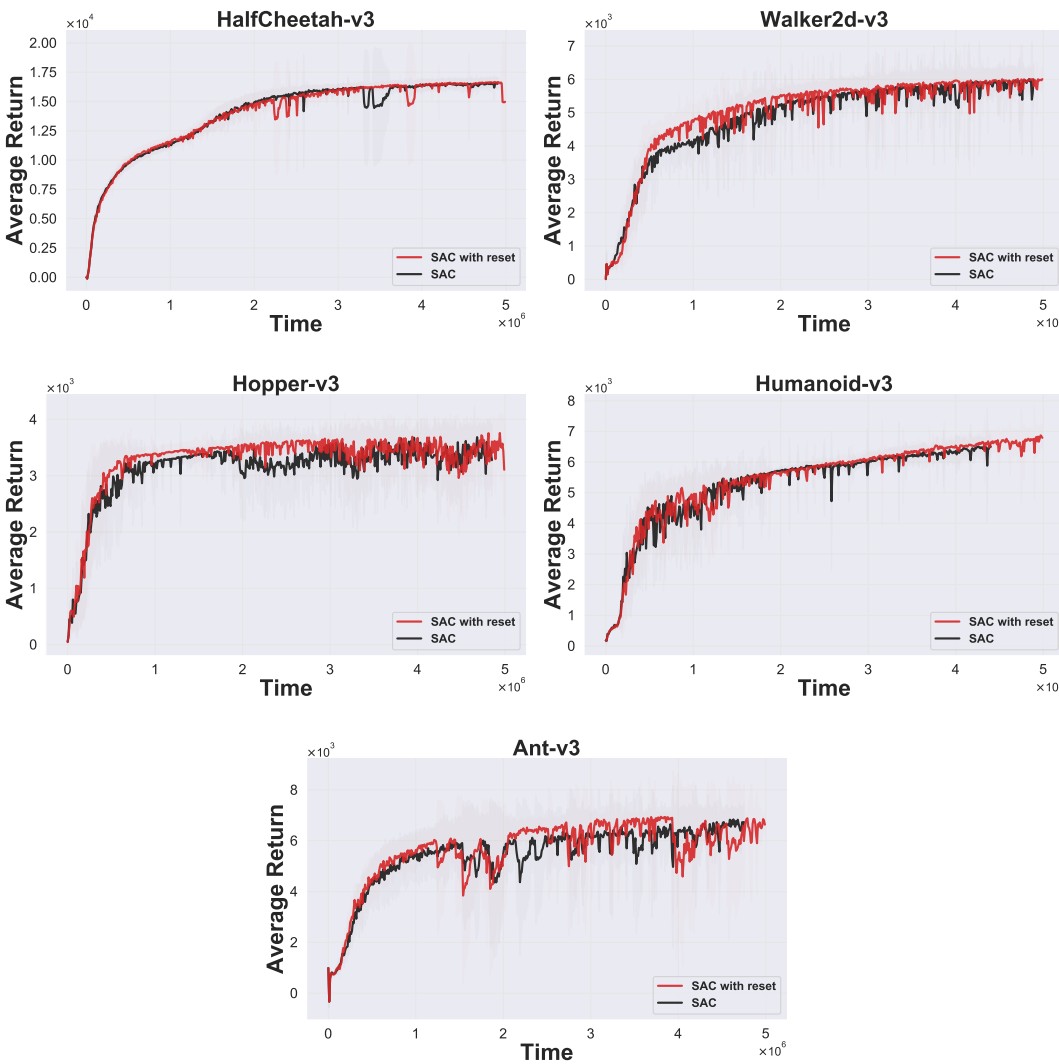

Figure 52: A comparison between Soft Actor-Critic (SAC) with and without resetting Adam on the standard MuJoCo tasks. In this study, both the actor and critic optimizers are reset every 5000 steps. The results are averaged over 10 different seeds.

