# OpenReview forum: "Resetting the Optimizer in Deep RL: An Empirical Study"
_NeurIPS.cc/2023/Conference — NeurIPS 2023 poster_

### Official Review · Reviewer_aUSe · 2023-06-17

**Soundness:** 3 good
**Presentation:** 3 good
**Contribution:** 3 good
**Rating:** 6
**Confidence:** 4

**Summary:**

The authors argue that Adam's internal parameters should be reset with each iteration. The authors demonstrate the effectiveness of this approach in the Atari domain.

**Strengths:**

- Results are convincing for Rainbow.
- Novelty is very low, but potential impact is high, if the result generalizes, there is little reason not to use this method in every DQN-style RL algorithm.

**Weaknesses:**

- As mentioned by the authors, novelty is low compared to Bengio et al.
- Results are only shown on Rainbow and do not appear to work for SAC (with reason) -- but does raise the question if the method is effective for other RL methods.
- There is limited insight. Can the authors show that initializing Adam's parameters with 0 is better than using the parameters from the previous iteration in a more concrete way? Such as examining the behavior of the actual values. The fact that not resetting is seemingly better at low values of K suggests that not resetting can provide a reasonable initialization for the parameters of Adam.

Minor
- The y-axis is unlabelled in several figures.

**Questions:**

As mentioned in weaknesses:
- Does this result generalize to other methods besides Rainbow? Such as DQN or more modern deep RL methods.
- Can the authors show that initializing Adam's parameters with 0 is better than using the parameters from the previous iteration in a more concrete way? Such as examining the behavior of the actual values.


**Limitations:**

No concerns.

---

> ### Author Rebuttal · Authors · 2023-08-09
>
> We appreciate the reviewer's reading our paper as well as the overall quite positive assessment. Thanks for pointing out that the paper's potential impact is high.
>
> - As mentioned by the authors, novelty is low compared to Bengio et al.
>
> Note that Bengio et al. proposed an approach to account for the staleness of the moment estimates in RL. Their approach is one that requires computing the Hessian, and so, it is not the most practical approach. Also, they did not conduct any experiment on the Atari domain with various algorithms and optimizers. We did not say that our novelty is low compared to them, merely that they also identified the same issue with using momentum-based optimizers in deep RL.
>
> - Results are only shown on Rainbow and do not appear to work for SAC (with reason) -- but does raise the question if the method is effective for other RL methods.
>
> As we have shown in the original experiments, resetting is useful for 1) Rainbow with Adam, 2) Rainbow with RMSProp, 3) Rainbow Pro with Adam 4) Rainbow with Rectified Adam. Additionally we added new results for 5) DQN with Adam and 6) IQN with Adam. Overall the resetting approach has been useful in all variants of DQN-like algorithms we tested. As for approaches such as SAC, notice that in the continuous control tasks the value of $K$ used in algorithms such as SAC is pretty small (often as small as $1$). Therefore, resetting the optimizer after each outer iteration corresponds to not giving any time to the optimizer to compute the moment estimates. We think that the positive impact of resetting in all DQN-like algorithms is strong enough to warrant publication.
>
> - There is limited insight. Can the authors show that initializing Adam's parameters with 0 is better than using the parameters from the previous iteration in a more concrete way?
>
> We examined the behavior of gradient estimates and their cosine similarity relative to the Adam's first moment estimates. There is no meaningful correlation between the two, suggesting that not resetting corresponds with arbitrarily initializing Adam estimates. Please see the attached pdf for the new results.
>
> - The y-axis is unlabelled in several figures.
>
> Thanks for catching this issue. We have labeled the axis.
>
> - Does this result generalize to other methods besides Rainbow? Such as DQN or more modern deep RL methods.
>
> Yes, they indeed do, based on our old and new experiments where we add results for DQN (and IQN) per your suggestion. Please see the pdf, and we hope the reviewer takes into account the added result in their final evaluation.

---

> > ### Comment · Reviewer_aUSe · 2023-08-16
> >
> > Thanks for the response and the additional results.
> >
> > At this time I don’t intend on increasing my score but continue to favor acceptance of the paper.

---

### Official Review · Reviewer_59Jp · 2023-06-30

**Soundness:** 2 fair
**Presentation:** 3 good
**Contribution:** 2 fair
**Rating:** 5
**Confidence:** 4

**Summary:**

The paper addresses the issue of using modern optimizers, such as Adam, which maintain internal parameters that are updated over time, potentially contaminating the optimization process. To mitigate this effect, the paper proposes a simple strategy of resetting the internal parameters of the optimizer at the start of each iteration. Empirical investigations using different optimizers and the Rainbow algorithm show that this modification enhances the performance of deep reinforcement learning on the Atari benchmark.

**Strengths:**

### Writing
The authors effectively communicate their ideas and concepts, ensuring clarity and coherence throughout the paper. The logical structure and well-reasoned arguments contribute to the overall quality of the essay. The article excels in providing the reader with a clear understanding of the problem's context and significance. By effectively conveying the goals and challenges of the study, the authors enhance the reader's comprehension of the subsequent experiments. Overall, the writing is of high quality, facilitating a smooth and engaging reading experience.

### Method
The paper presents an easy-to-use approach by introducing a method that is not only easy to implement, but also easy to apply, which enhances the potential adoption and practicality of the proposed approach.This user-friendly feature makes the method highly accessible and beneficial to researchers and practitioners in various fields.
The used code bases and hyperparameters are provided, allowing the results to be reproduced.

**Weaknesses:**

While I appreciate the proposed method's ease of use, I believe that the authors could have conducted a more comprehensive and statistically rigorous analysis of their approach, considering its simplicity.
One notable limitation of the paper is the absence of confidence interval plots and statistical analysis, which could have been derived from [1], to enhance the clarity and precision of the findings. Incorporating these elements would have allowed readers to better understand the level of uncertainty associated with the reported results, thus bolstering the overall robustness of the study.
Furthermore, the authors only rely on a single seed for the initial analysis, without providing a compelling rationale for this choice. Although using a single seed can streamline the experimental process, it diminishes the validity of the findings by disregarding potential result variations arising from multiple seeds. A more thorough explanation or a comparison of outcomes based on different seeds would have added value to the introduction, ensuring a more comprehensive analysis.
I appreciate that the authors included continuous control tasks in their study; however, these tasks are not thoroughly explored. While the authors provide hypotheses to explain the unexpected results, a deeper analysis would have been expected.

In line 293, the authors reference a follow-up paper on resetting approaches but fail to cite the original work [2], which states in the section "What and how to reset" that resetting the optimizer has almost no significant impact due to quick updates of the moments. This contradicts the findings of this work.
Which brings me to the conclusion that I believe that the paper shows promise and the authors have taken a positive direction. However, in its current form, the paper falls short of being acceptable. It is essential to include comparisons to other baselines, such as [2], to provide a more thorough understanding of the opportunities and limitations, and to gain a clearer understanding of the internal effects in order to explain the aforementioned points.

### Minor
- The protocol for the random resets is not easy to understand and should be specified more clearly

[1] Agarwal, Rishabh, et al. "Deep reinforcement learning at the edge of the statistical precipice." Advances in neural information processing systems 34 (2021): 29304-29320.
[2] Nikishin, Evgenii, et al. "The primacy bias in deep reinforcement learning." International Conference on Machine Learning. PMLR, 2022.


**Questions:**

- How does this method compare to other resetting approaches in terms of effectiveness?
- What is the level of statistical significance observed in the results?
- Why is resetting not effective for continuous control tasks?
- Are there any experiments demonstrating the impact of contamination on the tasks discussed in this paper?
- What are the consequences of reducing the frequency of optimizer resets beyond K=8000?
- How can an optimal value for $K$ be determined?
- Which ADAM/optimizer parameters are relevant when performing resets, i.e. have an effect when reset?
- Are the observed effects still present when modifying the ADAM/optimizer hyperparameters?
- Do different loss functions used in various DQN versions (e.g., MSE, Huber, Quantile) exhibit similar behaviors?

**Limitations:**

The authors have made some effort to address the limitations; however, it is crucial for them to conduct a more comprehensive investigation into these limitations, as mentioned in the "Weakness" section.

---

> ### Author Rebuttal · Authors · 2023-08-09
>
> We first would like to thank the reviewer for mentioning that our paper is high quality, facilitating a smooth and engaging reading experience. We also appreciate that the reviewer provided a detailed review.
>
> - While I appreciate the proposed method's ease of use, I believe that the authors could have conducted a more comprehensive and statistically rigorous analysis of their approach, considering its simplicity
>
> We agree that we should have ensured we get statistically significant results not just for the experiment with all 55 Atari games, but also with the ablation studies. Please see the attached pdf where we have added 9 new seeds to all ablation studies.
>
> - In line 293, the authors reference a follow-up paper on resetting approaches but fail to cite the original work [2], which states in the section "What and how to reset" that resetting the optimizer has almost no significant impact due to quick updates of the moments. This contradicts the findings of this work.
>
> We actually reference the paper [2] in our paper. It is our reference [47]. We are not sure why the reviewer is assessing our work negatively given that we actually cite this paper. We are happy to add more discussion about situating our work relative to this innovative paper.
>
> To summarize, notice that the motivation of their work and the techniques used for resetting are quite different. Nikishin et al. study the primacy bias phenomenon whereby the RL agent usually overfits to the training examples it encounters early during training, and thus can lose its plasticity. This is quite orthogonal to our motivation where we argue that an RL agent is facing a sequence of optimization problems and therefore we should reset the optimizer at the beginning of each new problem. Nikishin et al., reset parts of the agent's weights to maintain plasticity, whereas we discuss resetting the optimizer to ensure we do not contaminate the gradient steps taken by the optimizer. Finally, Nikishin et al. is conducting their experiments in the simpler continuous control setting with policy-gradient approaches where Polyak-based updates are often used for the target network. This is another significant deviation from our setting where we focus on the DQN style of algorithms where hard target-network updates are often preferred to the Polyak update. Overall, we have different motivations, techniques, and settings relative to Nikishin et al. We are happy to further clarify this in the text.
>
> - Why is resetting not effective for continuous control tasks?
>
> In the continuous control task the value of $K$ used in algorithms such as SAC and TD3 is pretty small (often as small as $K=1$). Therefore, resetting the optimizer after each outer iteration corresponds to not giving any time to the optimizer to compute the moment estimates. Clearly suppressing the optimizer in terms of not allowing it to compute moments is not a good idea.
>
> - Are there any experiments demonstrating the impact of contamination on the tasks discussed in this paper?
>
> Please see the attached pdf where we verified the contamination effect empirically.
>
> - What are the consequences of reducing the frequency of optimizer resets beyond $K=8000$?
>
> We are not sure what the reviewer is asking when they say ``reducing beyond $8000$''. If the question is about increasing this value beyond $8000$, in our experience, spending too much time solving each iteration is not a good idea since after each step of optimization, by standard convention, we also take a step in the environment. Therefore, it is usually better to crudely solve each iteration (smaller $K$) and move to the next one. Resetting allows us to move to the next iteration faster, which is why for resetting a smaller than $K=8000$ is usually a better value.
>
> - How can an optimal value for $K$ be determined?
>
> We would like to remind that we did not introduce this parameter. Therefore, we do not think it is incumbent upon us to answer this question in our paper as this is simply beyond the scope of our work. Also, not knowing the optimal $K$ is not at all reflective of a weakness of the resetting strategy, because we also need to tune $K$ when not resetting the optimizer. That said, this is a good empirical question, one that we would be super excited to focus on after publishing this paper.
>
> - Which ADAM/optimizer parameters are relevant when performing resets, i.e., have an effect when reset?
>
> We have noticed that we need to reset both Adam's first and second estimates to achieve the best performance. Only resetting the first or the second moment is not effective. We are happy to add this experiment to the paper.
>
> - Are the observed effects still present when modifying the ADAM/optimizer hyperparameters?
>
> The reviewer can find additional experiments in the original submission where we changed the Adam optimizer to RMSProp and Rectified Adam, and we observed that resetting is useful in those contexts too. Moreover, for the rebuttal we also added experiments on DQN and IQN, where again we showed that resetting is helpful. See the attached pdf for the new results.
>
> - Do different loss functions used in various DQN versions (e.g., MSE, Huber, Quantile) exhibit similar behaviors?
>
> Again, see our answer above. Rainbow, Rainbow Pro, IQN, and DQN are all using different loss functions. Nonetheless resetting is helpful in all cases above.

---

> > ### Comment · Reviewer_59Jp · 2023-08-11
> > **Thank you for your answer**
> >
> > For some reason I misread the citation [47] and thought you were referring to this follow-up paper:
> > D'Oro, P., Schwarzer, M., Nikishin, E., Bacon, P. L., Bellemare, M. G., & Courville, A. (2023). Sample-Efficient Reinforcement Learning by Breaking the Replay Ratio Barrier. In The Eleventh International Conference on Learning Representations.
> > Please apologize and disregard this comment.
> >
> > I also appreciate the clarification of why Nikishin et al. come to a different conclusion about the optimizer reset, which at the same time gives some intuition about the results for the continuous control setting.
> >
> > >  What are the consequences of reducing the frequency of optimizer resets beyond ?
> >
> > This question was probably badly worded. But as guessed correctly, I meant an increase of $K$, which implies a decrease in the frequency of optimizer resets.
> >
> > > We would like to remind that we did not introduce this parameter. Therefore, we do not think it is incumbent upon us to answer this question in our paper as this is simply beyond the scope of our work. Also, not knowing the optimal is not at all reflective of a weakness of the resetting strategy, because we also need to tune when not resetting the optimizer. That said, this is a good empirical question, one that we would be super excited to focus on after publishing this paper.
> >
> > Thank you for the answer.  With this question, I did not mean to imply that choosing an appropriate $K$ is a weakness of the proposed method, since, as you mentioned, it has to be done regardless. I was just interested in how the added purpose of $K$ for the optimizer resets affects the hyperparameter selection of $K$.
> >
> > > Are the observed effects still present when modifying the ADAM/optimizer hyperparameters?
> >
> > This question was more related to changing the hyperparameters of the respective optimizers than to the optimizer itself, e.g. the betas for ADAM. Granted, the default values of 0.9, 0.999 in the case of ADAM are usually the best settings. I would just be curious if you have tried other values besides the default and possibly seen a difference.
> >
> > Other than that, I appreciate the effort that went into the rebuttal answers as well as the additional results and seeds. They definitely further support the proposed approach. I will take this into account when discussing the final recommendation with the other reviewers and AC.

---

> > > ### Author Response · Authors · 2023-08-11
> > > **Changing Adam's Hyperparameters**
> > >
> > > In terms of the Adam optimizer, we used the default values of Adam because deviating from the default values could have been interpreted as our trying too hard to show the effectiveness of resetting. In particular, other than $\beta_1$ and $\beta_2$, we also did not change the default step-size value of Adam used in baseline RL algorithms to keep the comparison as grounded as possible relative to the literature.
> > >
> > > That said, we totally agree that it would be interesting to look at the effects of these hyper-parameters on the validity of our conclusions. For example, the moment estimates may become less or more stale depending on the choices of $\beta_1$ and $\beta_2$, and this may ultimately affect the gap between the resetting and non-resetting agents. However, in light of our lack of ability to edit the rebuttal pdf, it seems impossible for us to carry this experiment, so unfortunately we can only promise to add this experiment in our final paper.
> > >
> > > We are delighted to see the reviewer's engagement with our rebuttal, as well as the reviewer's stating that they will take into account the added seeds, the contamination-measurement experiment, and the IQN/DQN results in their final recommendation. However, we also worry that our average score has remained low, and that this can negatively affect our chances in the discussion. While we appreciate if the new considerations could also be reflected in the final score, we nevertheless really respect the reviewer's engagement with our rebuttal.

---

> > > > ### Comment · Reviewer_59Jp · 2023-08-17
> > > > **Thank you for additional details.**
> > > >
> > > > Thank you for providing some more insight on that matter. I have updated the score of the original proposal.

---

> > > > > ### Author Response · Authors · 2023-08-17
> > > > > **Thanks**
> > > > >
> > > > > We appreciate the reviewer for their reconsideration and for supporting our paper. Please let us know if there was any new or lingering question.

---

### Official Review · Reviewer_5d1P · 2023-07-03

**Soundness:** 1 poor
**Presentation:** 3 good
**Contribution:** 3 good
**Rating:** 6
**Confidence:** 4

**Summary:**

This paper questions the standard use of Adam-type optimizers in deep RL. The paper argues that solution methods in deep RL are best thought of as solving a sequence of optimization problems. And that the standard use of optimizers leads to "contamination" of the optimizer's internal parameters. The paper then proposes to reset the optimizer's internal parameters to fix this "contamination." Finally, the experiments on Atari show that resetting the optimizer's internal parameters leads to significant performance improvement.

**Strengths:**

The main strength of the paper is that the key idea of the paper, resetting optimizer parameters at the beginning of each iteration, is simple and effective. I liked the general theme of the paper, i.e., we need to understand better the tools we borrow from other fields. The paper is well-written and easy to understand, making it accessible to a wide audience. The experiments in section 4.3 are performed on 55 Atarti games with ten seeds each, which might mean the results are statistically significant. Resetting seems to be beneficial for multiple optimizers like RMSprop, Adam and Rectified Adam.

**Weaknesses:**

The paper has two major weaknesses:
1. The paper claims at many points that a "contamination" effect plagues RL (for example, lines 107-109) and that many updates are wasted to unlearn the effects of the previous iteration. However, the paper does not describe what exactly this "contamination" means, and neither does it show the presence of any "contamination." All the paper shows is that there is a performance boost when we reset the optimizer's internal parameters. This performance boost is not direct evidence of contamination from iteration to iteration.

However, this weakness can be easily overcome. The paper first needs to contain a definition of "contamination," maybe the authors mean that the internal parameters($m$ and $v$) are too far away from their true values at the beginning of each iteration. One way to measure this difference could be to measure the cosine similarity between the current value of $m$ and the true value of $m$. The true value can be measured by taking the gradient of all the samples in the buffer and taking steps using that gradient. A large difference between the true and current value of $m$ would mean contamination. The paper also suggests that this contamination is particularly large at the beginning of each iteration compared to a random time in the learning process. Again, this can be easily shown by showing that the difference in true $m$ and current $m$ is larger at the beginning of the iteration compared to any random time in the learning process.

2. None of the results in the paper except the ones in section 4.3 are statistically significant. The paper only shows results for a single random seed. I should note that the authors are aware of this weakness (line 138). The best way to look at the current experiments in Sections 4.1 and 4.2 is that they are used to tune hyper-parameters for the experiments in Section 4.3. The authors might be limited in their computational resources, but in that case, it is better to present statistically significant results in smaller environments like MinAtar[1] than unreplicable results in a big environment.

Other than these two main problems, there are a few other minor issues in the paper.
1. The update equations for Adam in Section 3 are wrong. Instead of using $m$ and $v$ for the final update, new variables $\hat{m}$ and $\hat{v}$ are used. See to the original Adam paper for the correct equations.
2. Line 176 says that $K=1$ corresponds to vanilla gradient descent. But, that is not true. For $K=1$, the update is similar to the Rprop optimizer, not SGD. For $K=1$, the update only takes into account the sign of the partial derivative but not its magnitude.
3. The value of $K$ is not properly tuned. In Figure 4, the difference between $K=1000$ and $K=500$ is insignificant. So, the optimal value of $K$ could be smaller. I suggest the authors also try smaller values of K, like 250, 125, etc.

I like the ideas presented in the paper. However, I can not recommend accepting the paper in its current form in light of these weaknesses.

[1] Young, K., & Tian, T. (2019). Minatar: An atari-inspired testbed for thorough and reproducible reinforcement learning experiments. arXiv preprint arXiv:1903.03176.


**Questions:**

What would be a good definition of "contamination"?

**Limitations:**

No confidence intervals are reported for any experiment in the paper. I recommend the authors report the 95% bootstrapped confidence intervals for their results. [2] and [3] provide good guidelines for properly reporting experimental results in deep RL.


[2] Agarwal, R., Schwarzer, M., Castro, P. S., Courville, A. C., & Bellemare, M. (2021). Deep reinforcement learning at the edge of the statistical precipice. Advances in neural information processing systems, 34, 29304-29320.
[3] Patterson, A., Neumann, S., White, M., & White, A. (2023). Empirical Design in Reinforcement Learning. arXiv preprint arXiv:2304.01315.

EDIT:

I have updated my score based on the new results provided by the authors.

---

> ### Author Rebuttal · Authors · 2023-08-09
>
> We thank the reviewer appreciating our idea. In what follows, we address the particular weaknesses and questions raised.
>
> - The paper does not describe what exactly this contamination means, and neither does it show the presence of any contamination.
>
> Please take a look at the new cosine similarity experiment in the attached pdf. We see very little cosine similarity between the gradient and Adam's moment estimate immediately after moving into a new iteration. If the moment estimate was not stale, you would expect them to form a low angle with the gradient, and therefore give us high cosine similarity, but this is demonstrably not the case for all games we tested.
>
> Thanks for your suggestion in conducting the cosine similarity experiment, though notice that computing the true value of m like you suggested is extremely expensive, and so inspired by your comment, we conducted a similar experiment where we measured the cosine similarity between the Adam's first moment estimates and the stochastic gradient computed immediately after moving to the next iteration.
>
> - None of the results in the paper except the ones in section 4.3 are statistically significant.
>
> That is a fair criticism, and in light of your concern, we updated all of our results to include 10 random seeds. We hope that the reviewer considers increasing their score since they stated using 1 random seed as one of the main weaknesses of our work. To be fully transparent, we were limited by compute at the time of submission and made the strategical decision to use most compute for the experiment with all 55 Atari games.
>
> - The update equations for Adam in Section 3 are wrong.
>
> You are correct, but we also want to reassure the reviewer that we know the Adam update. We just miss $\hat m$ and $\hat v$. This was simply a typo which we have addressed in the main text. Thanks for your comment nonetheless.
>
> - Line 176 says that  corresponds to vanilla gradient descent.
>
> You are correct, and we will fix that.
> - No confidence intervals are reported for any experiment in the paper.
>
> Please see the attached pdf for the updated results. We now report confidence intervals so we hope the reviewer reconsiders their evaluation.
>
> -  I suggest the authors also try smaller values of K, like 250, 125, etc.
>
> This is a good suggestion and we are happy to add this in the final paper if accepted. For the rebuttal we prioritized making the existing experiments statistically significant. We really hope the reviewer considers increasing their score and help us get the paper to the finish line and we will be happy to then use our time to try values of K between 500 and 1.
>
> - What would be a good definition of ``contamination''?
>
> A definition we have adopted is small cosine similarity between the first moment estimate and the gradient vector immediately after updating the target network. If the two vectors are very rarely aligned, the moment estimate is in fact just misleading the agent, and so we call this a contamination effect.

---

> > ### Comment · Reviewer_5d1P · 2023-08-16
> >
> > I thank the authors for adding more seeds and experiments to measure contamination.
> >
> > The results with more seeds are largely consistent with the results in the original manuscript. The new results alleviate most of my concerns about the statistical significance of the results in the paper.
> >
> > The new contamination results are striking. The cosine similarity effectively reduces to zero after about 20M frames. These results seem to support the claim that there is contamination. However, they are not fully satisfactory. The authors approximate the true gradient from the first mini-batch in the next iteration. But I don't know if the first sample gradient is a good approximation of the true gradient. The true value is the average over a sample of all the data in the buffer (1M?), but the mini-batch is only 128(?). I realize that computing the gradient over the full buffer will be computationally expensive. Still, we can reach a middle ground by measuring the average gradient for 10,000 or 50,000 transitions in the buffer.
> >
> > A few questions to the authors:
> > - Why does the cosine similarity decrease over time? And why was it exactly 1 in the beginning? If there is contamination from iteration to iteration, shouldn't the cosine similarity always be zero? Why do you think that is the case?
> > - What do the error bars report? Is it the 95% bootstrapped confidence interval or IQM or something else?
> > - Did you get to do more runs for rainbow without resetting (Figure 4) for K=0? I'm very curious to see if Rainbow with K=0 and K=8000 perform the same. If that is the case, it would mean that we can effectively remove the target network from Rainbow.
> >
> > As a side note, the authors do not describe the hyper-parameters used for all the algorithms in the paper. The authors should add a section in the appendix describing all the hyper-parameters for all the algorithms used in the paper.
> >
> > I light of the authors adding more runs to make their results statistically significant and adding a measure for contamination, I have changed my score to accept the paper.

---

> > > ### Author Response · Authors · 2023-08-17
> > > **Answering Additional Questions**
> > >
> > > We thank the reviewer for their continued engagement with our rebuttal, and for their neat suggestion about the contamination experiment which truly strengthened our paper.
> > >
> > > - The true value is the average over a sample of all the data in the buffer (1M?), but
> > > the mini-batch is only 128(?).
> > >
> > > You are correct in saying that the gradient is noisy because it is only estimated over a minibatch of size 64, however, we think the low cosine similarity is not due to the fact that we use stochastic gradients and it is due to the fact that the optimization landscape actually changes from one iteration to the next one. That said, to further address the reviewer's concern, we just started another experiment where we change the batch size from 64 to 2048, and will add this to the paper. However, please keep in mind that even if a much larger batch size can increase cosine similarity, in practice we do not want to use a larger batch size anyways due to 1) computational considerations and 2) that in deep learning very large batch sizes actually performs poorly due to their converging into sharp minimizers with poor generalization gap (Keskar et al).
> > >
> > > - Why does the cosine similarity decrease over time? And why was it exactly 1 in the beginning? If there is contamination from iteration to iteration, shouldn't the cosine similarity always be zero? Why do you think that is the case?
> > >
> > > This is a very good question. We think the early cosine similarity we observe is due to the fact the network is initialized completely randomly, and so the optimizer roughly moves the weights in the same direction irregardless. However, this is a very quick and transient phase and the cosine similarity quickly vanishes to zero.
> > >
> > > - What do the error bars report?
> > >
> > > Correct, it is the 95\% bootstrapped confidence intervals.

---

> > > > ### Author Response · Authors · 2023-08-17
> > > > **Answering Additional Questions (2)**
> > > >
> > > > - Did you get to do more runs for rainbow without resetting (Figure 4) for K=0? I'm very curious to see if Rainbow with K=0 and K=8000 perform the same. If that is the case, it would mean that we can effectively remove the target network from Rainbow.
> > > >
> > > > Yes we did, and we still observed that Rainbow (without resetting) performs equally well with the extreme value of $K=1$. In fact, we see that the performance of Rainbow remains flat with respect to $K$. In our experience, only when using extremely large values of $K$ (bigger than 20000) we get a negative impact on Rainbow's performance. We concede that this is a somewhat surprising result, but note that in the literature we have seen examples where a much smaller value of the original $K=8000$ is used for Rainbow. For example, Agent 57 used $K=1500$ (Badia et al). Overall, akin to the reviewer, we are also really surprised that Rainbow works equally well with $K=1$ and that the choice of $K=8000$ seems to just be a legacy from previous papers such as DQN. The reviewer correctly identifies that by removing the frozen target network in Rainbow without resetting we will not affect the performance negatively.
> > > >
> > > >
> > > > - As a side note, the authors do not describe the hyper-parameters used for all the algorithms
> > > >
> > > > We will be sure to add a section on hyper-parameters. Note, however, that we have used the standard hyper-parameters from the Dopamine implementation (Castro et al). The main hyper-parameter we worked with was $K$, which we have reported in the paper. That said we agree that having a hyper-parameter section would make the paper self-contained, and we will add this to the paper.
> > > >
> > > > Again, thanks for your constructive feedback. It really helped improve our paper. Please let us know if any question has lingered at this point.
> > > >
> > > > Keskar et al, ''On Large-Batch Training for Deep Learning: Generalization Gap and Sharp Minima", 2017
> > > >
> > > > Badia et al, ''Agent57: Outperforming the Atari Human Benchmark", 2020
> > > >
> > > > Castro et al, ''Dopamine: A research framework for deep reinforcement learning", 2018

---

> > > > > ### Comment · Reviewer_5d1P · 2023-08-17
> > > > >
> > > > > Dear Authors, thank you for quickly answering my questions. I want to follow up on a couple of points.
> > > > >
> > > > > I want to clarify that when I suggest using a larger sample (say 10k) to estimate the true gradient, I'm not suggesting running the entire experiment with a 10k mini-batch size. I'm suggesting using a mini-batch of size 64 and, at transitions, passing a mini-batch of 10k through the network to get an estimate of the true gradient but do not use that estimate to update the weights as that will change the underlying algorithm.
> > > > > The result showing that removing the target network from Rainbow does not hurt the performance is very exciting. I think the authors should spend some paragraphs in the paper and maybe a couple of lines in the conclusion pointing out this result. I realize this is not the paper's main message, but this result will be an important data point for future research.

---

> > > > > > ### Author Response · Authors · 2023-08-17
> > > > > > **Rainbow's performance with K=1**
> > > > > >
> > > > > > We definitely agree that expanding the discussion about Rainbow without resetting and $K=1$ can really be helpful for future research, especially the kind of research that aims to better understand the frozen target network, or to potentially even remove it altogether! In particular, we also have found that, unlike Rainbow, DQN really struggles with small values of $K$ and works terribly with $K=1$. While this explains why the original $K=8000$ choice was "grandfathered in" from DQN to Rainbow, it seems to us that one (or more) of the components of Rainbow is obviating the need to use larger $K$. We don't have a definitive answer yet, and so this should be further investigated in future.
> > > > > >
> > > > > > Also, like you said, we did not focus on this result too much in the submission because it was only tangentially related to the main message of our paper. We really hope the reviewer would reinforce their advocacy of our paper so we could present the surprising Rainbow($K=1$) result in the final camera ready.
> > > > > >
> > > > > > As for larger batch size, yes, we agree and understood the reviewer's intent. Please stay tuned and we will get back to you on the final outcome of the experiment. It seems like we can not edit the pdf and also cannot provide outside links, but we will state what we observe here as a comment.

---

> > > > > > > ### Author Response · Authors · 2023-08-20
> > > > > > >
> > > > > > > We thank the reviewer for their continued engagement and for advocating our work.
> > > > > > >
> > > > > > > Per your request, we ran the experiment with a larger batch size and measured the cosine similarity after target updates on all 12 games from our ablation study. We show results averaged over all games for the original batch size (64) and the much larger batch size (2048).
> > > > > > >
> > > > > > > | number of frames | cosine similarity (64) | cosine similarity (2048) |
> > > > > > > | ------ | ------ | ---- |
> > > > > > > | 0 M      | 0.982 $\pm$ 0.1374      | 0.9884 $\pm$ 0.0802   |
> > > > > > > | 10 M     | 0.1034 $\pm$ 0.2808      | 0.1872 $\pm$ 0.2652    |
> > > > > > > | 20 M      |0.0239 $\pm$ 0.1052      | 0.0259 $\pm$ 0.0655  |
> > > > > > > | 30 M      | -0.0085 $\pm$ 0.0842     |0.04 $\pm$ 0.0683   |
> > > > > > > | 40 M      | -0.0039 $\pm$ 0.103     | 0.0122 $\pm$ 0.0746   |
> > > > > > > | 50 M      | -0.0132 $\pm$ 0.0684    | -0.0154 $\pm$ 0.0871   |
> > > > > > > | 60 M      | -0.0121 $\pm$ 0.0751     |0.0339 $\pm$ 0.0658  |
> > > > > > >
> > > > > > > Overall, under both batch sizes we see a similar trend where the cosine similarity goes to zero. We also conducted a related experiment where we measured the cosine similarity right before target updates. Please refer to our last comment to reviewer Ks5C.

---

### Official Review · Reviewer_Ks5C · 2023-07-06

**Soundness:** 3 good
**Presentation:** 3 good
**Contribution:** 2 fair
**Rating:** 6
**Confidence:** 4

**Summary:**

The paper studies optimization in value-based deep reinforcement learning. The key insight is that when using target networks for action-value function training, changes in the target parameters yield a change in the optimization problem the online parameters are solving. Because of that, the authors argue that preserving the adaptive optimizer statistics (e.g. of Adam) might or might not be desirable. The paper then studies the effect of resetting the optimizer state after (hard) target updates mostly using the Rainbow algorithm on Atari games as a testbed yielding a slight positive aggregate improvement.

**Strengths:**

The main strength of the paper is the simplicity of the contribution; the paper is well-written and easy to follow, and the method is motivated and described well. The experimental protocol is solid: it uses the full set of 55 Atari games and a standard Rainbow implementation.

**Weaknesses:**

The main weakness of the paper is the mixed empirical results. Granted, the median human-normalized performance improves from ~1.75 to ~2.25, however, per-game effects from resetting the optimizer are highly heterogeneous, yielding performance deterioration in ~14 environments. The soundness of the paper could have been higher if, at least, an explanation (supported by evidence) for the negative effects was given.

**Questions:**

Target network parameter updates indeed change the loss landscape that the online parameters are navigating. In addition to that, updating the replay buffer changes the distribution of inputs and hence the optimization problem for online parameters. Do you have ideas on how an optimizer could be changed to adapt to the input shifts?

**Limitations:**

“We hypothesize that this can contaminate the internal parameters of the employed optimizer in situations where the optimization landscape of the previous iterations is quite different from the current iteration.” (L9) The reviewer didn’t find an empirical verification of this assumption.

Many deep RL algorithms use moving average target updates after each step instead of periodic hard updates. The authors demonstrate preliminary evidence that in soft actor-critic that uses such a practice, the optimizer resets do not improve the performance. Having said that, the reviewer appreciates the transparency about the negative results.

Again, one of the limitations is that in some environments resetting the optimizer yields negative results. It implies that a better alternative could be triggering the optimizer reset using a criterion (e.g. based on a measure of the loss landscape change / by performing a lookahead and assessing whether the reset was helpful)

---

> ### Author Rebuttal · Authors · 2023-08-09
>
> We thank the reviewer for the time spent carefully reviewing the paper and for appreciating our work. Please find below some clarification regarding your questions.
>
> - "Granted, the median human-normalized performance improves from 1.75 to 2.25, however, per-game effects from resetting the optimizer are highly heterogeneous, yielding performance deterioration in 14 environments."
>
> We ask the reviewer to consider the fact that resetting the optimizer is an extremely simple modification of the original algorithm in that it is adding no new hyper-parameter and/or computation to the baselines. We believe that due to the simplicity of resetting and the fact that it is a very natural thing to do, the bar should not be set very high to the point where we expect dramatic improvement on all games. In order words, simplicity of the approach is in our view a strength of the approach and not a bug, and so the improvements should be celebrated more given this simplicity.
>
> Also, by looking at the original Rainbow paper, we observe that the positive impact due to each individual component of Rainbow is not that high, and so in light of that, we are actually positively surprised that the simple resetting idea can be so effective. We did not explore adaptive versions of resetting that could have been even more dominant relative to not resetting, but we think the fact that the simple resetting is so effective can instigate future work that can come up with even more performant resetting strategies.
>
> - In addition to that, updating the replay buffer changes the distribution of inputs and hence the optimization problem for online parameters. Do you have ideas on how an optimizer could be changed to adapt to the input shifts?
>
> This is a very good point, and we indeed agree that the online RL problem (in contrast to offline RL) is non-stationary even from one time-step to the other. We did not explore approaches that account for shifts in inputs as they were beyond the scope of this result. That said, the mere fact that the reviewer is, akin to us, interested in this question is a testament that our current results are insightful, thought-provoking, and would instigate further research in this direction.
>
> - ``We hypothesize that this can contaminate the internal parameters of the employed optimizer in situations where the optimization landscape of the previous iterations is quite different from the current iteration.'' (L9) The reviewer didn’t find an empirical verification of this assumption.
>
> Please take a look at the new cosine similarity experiment in the attached pdf. We see very little cosine similarity between the gradient and Adam's first moment estimate immediately after moving into a new iteration, which empirically verifies the contamination effect.

---

> > ### Comment · Reviewer_Ks5C · 2023-08-17
> >
> > Thanks for the response.
> >
> > I concur that it is encouraging to see aggregate improvements from such a simple, hyper-parameter-free technique. What I was mostly pointing at was a lack of explanation for why some of the games experience worse performance from applying optimizer resets.
> >
> > The cosine similarity experiment is indeed interesting. However, it seems like it is unable to explain the effects of resetting the optimizer. For instance, based on Figure 8, DemonAttack is an example of a game with positive effects from opt-reset and Breakout is an example with neutral effects. For both of these games the cosine similarity between the Adam 1st moment and the gradient after the target update behaves almost identically, suggesting that low cosine similarity isn't explaining the opt-reset effects. Additionally, for a complete picture, we need a similar plot _with cosine similarity right before the target update_.
> >
> > Thanks for also validating opt-reset on DQN and IQN — was the aggregate score calculated on all 55 games?

---

> > > ### Author Response · Authors · 2023-08-18
> > > **Further Clarifications**
> > >
> > > Thanks for your carefully reading the paper and the rebuttal.
> > >
> > > The reviewer makes a good observation. We acknowledge that measuring the cosine similarity alone cannot fully predict the utility of resetting. Note that in doing this experiment, our intention was merely to show that a contamination effect does exist, but even with different amounts of contamination it is still possible for the non-resetting agent to occasionally outperform the resetting agent due to the presence of other confounding factors in deep RL.
> > >
> > > To take a step back and focus on the bigger picture, in this work we presented the view that mainstream RL algorithms should better be thought of as solving for a sequence of optimization problems. This stands in contrast to the view that treats RL algorithms as solving for a single optimization problem. Fully adopting this view means that the most natural thing to do is to reset the RL optimizer in situations where the optimizer is moment-based. This surprisingly simple technique was somehow absent in the literature, and while we do not have all answers about when it works best, we observe that over many settings of RL algorithms and optimizers it is quite beneficial to reset. We believe that a paper that takes a first step to present this optimization view and to propose the resetting strategy can serve the RL literature well.
> > >
> > > - Thanks for also validating opt-reset on DQN and IQN — was the aggregate score calculated on all 55 games?
> > >
> > > To answer your question, we did the new 1) DQN with Adam and 2) IQN with Adam experiments only on 12 games. These are the same 12 games from section 4.1 used in the original submission to show the benefits of resetting on 3) Rainbow with Adam, 4) Rainbow Pro with Adam, 5) Rainbow with Rectified Adam, and 6) Rainbow with RMSProp. So taken together, we have 6 examples of resetting's improvement on the same set of 12 games. We chose these 12 games completely randomly when we began the project.
> > >
> > > Our experiments on the full 55 games are only on 1) Rainbow with Adam and 2) Rainbow with Rectified Adam.

---

> > > > ### Comment · Reviewer_Ks5C · 2023-08-18
> > > >
> > > > Could you give the plot with cosine similarities before the target updates?
> > > >
> > > > I'd also like to put on your radar the paper by Dabney et al. [1] that treats RL as a sequence of prediction problems.
> > > >
> > > > [1] Dabney, Will, André Barreto, Mark Rowland, Robert Dadashi, John Quan, Marc G. Bellemare, and David Silver. "The value-improvement path: Towards better representations for reinforcement learning." In Proceedings of the AAAI Conference on Artificial Intelligence, vol. 35, no. 8, pp. 7160-7168. 2021.

---

> > > > > ### Author Response · Authors · 2023-08-18
> > > > >
> > > > > Yes, we are working to report this experiment. We submitted it a few hours ago and because Rainbow takes roughly a day to complete 25 Million frames we expect to have a meaningful update in 2 or 3 days. Also notice that we cannot edit the pdf, and cannot provide links per the AC guidance, so we have to be creative in terms of how we report this result. Pardon us in advance for the inconvenience.
> > > > >
> > > > > - I'd also like to put on your radar the paper by Dabney et al. [1] that treats RL as a sequence of prediction problems.
> > > > >
> > > > > Yes, we are aware of the innovative work of Dabney et al. and we thankfully did not fail to cite it in the original submission (reference [30]). We also discuss it in the paper (lines 276 to 280). This paper is presenting a similar view that the RL agent should best be thought of as one that solves a sequence of value-prediction problems rather than a single one. Adopting this view led them to design their representation learner in a sequence-aware manner. More specifically, they argue that the representation learning aspect of the RL agent should aim to learn a representation that is well-equipped to predict, not just the current value function, but the sequence of value functions observed so far. They show that this will help learn better representations in the context of Atari.
> > > > >
> > > > > While Dabney et al. looks at this through the lens of representation learning, we were interested to explore the ramifications of this view through the lens of optimization. This led us to resetting which is the most natural thing to do having adopted the sequence view. We believe that adopting the sequence view can have interesting ramifications beyond optimization (our paper) and representation learning (Dabney et al.) as well. Even in the context of optimization, one can think of other ramifications, for example exploring different ways of initializing the optimization problem at each iteration in light of the sequence view, or looking at the previously solved optimization problems to better guide the agent in solving the current optimization problem. Our belief is that these are really exciting but under-explored questions, and we hope that by allowing us the opportunity to share our results with the community we can instigate further work in this direction. We are happy to add this discussion to the paper to better situate us relative to Dabney et al.

---

> > > > > > ### Author Response · Authors · 2023-08-20
> > > > > >
> > > > > > We thank the reviewer again for their engagement. Per your question, we measured cosine similarities after and before target network updates. We did so on the same 12 games we ran all our ablation studies on. See results below:
> > > > > >
> > > > > > In the first table we report the average cosine similarity across learning per each game.
> > > > > >
> > > > > >
> > > > > > | Game | average cosine similarity (after) | average cosine similarity (before) |
> > > > > > | ------ | ------ | ---- |
> > > > > > | Amidar      |0.0036$\pm$ 0.0021 | 0.4008$\pm$ 0.2396    |
> > > > > > | Asterix     |0.0025$\pm$ 0.0011 | 0.3941$\pm$ 0.146    |
> > > > > > | BeamRider      | 0.0059$\pm$ 0.0008 | 0.3763$\pm$ 0.1601   |
> > > > > > | Breakout      | 0.0021$\pm$ 0.0015 | 0.3622$\pm$ 0.1531   |
> > > > > > | CrazyClimber      |0.0074$\pm$ 0.0047 | 0.4034$\pm$ 0.2733   |
> > > > > > | DemonAttack      | 0.0003$\pm$ 0.0002 | 0.3972$\pm$ 0.1961   |
> > > > > > | Gopher |0.0069$\pm$ 0.0032 | 0.4139$\pm$ 0.1272 |
> > > > > > | Hero | 0.0048$\pm$ 0.0005 | 0.39$\pm$ 0.1471|
> > > > > > | Kangaroo | 0.0063$\pm$ 0.0052 | 0.3371$\pm$ 0.1605 |
> > > > > > | Phoenix |0.0042$\pm$ 0.0024 | 0.3996$\pm$ 0.1576|
> > > > > > | Seaquest | 0.0048$\pm$ 0.0014 | 0.4073$\pm$ 0.3206 |
> > > > > > | Zaxxon | 0.0045$\pm$ 0.0043 | 0.3933$\pm$ 0.1956 |
> > > > > >
> > > > > > In the second table we report cosine similarity during learning averaged over all 12 games:
> > > > > >
> > > > > > | number of frames | cosine similarity (after) | cosine similarity (before) |
> > > > > > | ------ | ------ | ---- |
> > > > > > | 0 M      | 0.982 $\pm$ 0.1374      | 0.9303 $\pm$ 0.1627    |
> > > > > > | 10 M     | 0.1034 $\pm$ 0.2808      | 0.4106 $\pm$ 0.2574    |
> > > > > > | 20 M      |0.0239 $\pm$ 0.1052      | 0.3493 $\pm$ 0.2074   |
> > > > > > | 30 M      | -0.0085 $\pm$ 0.0842     |0.4155 $\pm$ 0.1462   |
> > > > > > | 40 M      | -0.0039 $\pm$ 0.103     | 0.3148 $\pm$ 0.1537   |
> > > > > > | 50 M      | -0.0132 $\pm$ 0.0684    | 0.4046 $\pm$ 0.1222   |
> > > > > > | 60 M      | -0.0121 $\pm$ 0.0751     | 0.3366 $\pm$ 0.2077  |
> > > > > >
> > > > > > Overall, we see small cosine similarity after target updates, and we also see cosine values before target updates converging to values in the interval of [0.3, 04].

---

> > > > > > > ### Comment · Reviewer_Ks5C · 2023-08-20
> > > > > > >
> > > > > > > Thank you for the update! These experiments are insightful and I recommend including them into the paper. In response to providing evidence about the gradient contamination claim, I increase the score.

---

> > > > > > > > ### Author Response · Authors · 2023-08-20
> > > > > > > > **Thanks**
> > > > > > > >
> > > > > > > > We highly appreciated your insightful suggestion and your commitment to engage with our rebuttal. We will be sure to add these to the paper. Thanks for reinforcing your support.

---

### Author Rebuttal · Authors · 2023-08-09

We appreciate our reviewers for their thoughtful feedback. Despite their constructive criticism, we believe that all reviewers see positive aspects in our work. In particular, reviewer Ks5C agrees that our approach is motivated and well-described and that we ran our main experiments on all 55 Atari games with 10 seeds. Reviewer 5d1P agrees with the main message we liked to convey in this paper, namely that we need to understand better the tools we borrow from other fields, in this case deep-learning optimization. Reviewer 59Jp states that the paper presents an easy-to-use approach by introducing a method that is not only easy to implement, but also easy to apply, which enhances the potential adoption and practicality of the proposed approach, and finally Reviewer aUSe mentions that the impact of our work could be potentially high given the wide-spread use of modern optimizers in deep RL.

That said, Reviewers 5d1P and 59Jp raised the issue pertaining to using only one seed in some of our experiments. We first reiterate that for our main experiment on all 55 Atari games, we used 10 random seeds which is more than or at least on par with mainstream deep RL papers. Because of the volume of our ablation studies (already more than 1000 experiments) we were not able to use more than 1 seed in our ablations when submitting the paper. That said, we understand your concern, and agree that with the ablation studies, too, we need to have made sure that our results are statistically significant akin to what we did with the full experiment with all 55 Artari games.

To this end, we added 9 more seeds to the single seed we originally used for the ablation experiments, so in total we currently have 10 seeds for our ablation and main experiments. We updated Figures 1, 2, and 3 in the paper and we added error bars as requested by two of the reviewers. Please see the attached pdf for updated results. We are happy to add more seeds to all experiments in the final camera-ready version of the paper. We are under the impression that reviewers 5d1P and 59Jp overall liked the ideas presented in this work, but gave us a low score in large part due to the seed issue, so we really hope that they reconsider their score in light of now using 10 seeds in all experiments.

Moreover, reviewers 5d1P and 59Jp asked us to elaborate further on the contamination effect that we claim to have plagued modern optimizers in deep RL. To recap, we showed that TD/Rainbow/DQN/etc could be thought of as RL algorithms that solve a sequence of optimization problems:

$\theta^{t+1}\leftarrow \arg\min_{w} H(\theta^t,w)\ ,$

where we use Adam (without resetting) to approximately solve all iterations. Here, by contamination we mean that the Adam moment estimates computed in the previous iterations $(t-1, t-2, ...)$ correlate weakly (if at all) with the gradient of the objective function at iteration $t$. To support our claim, we conducted a new experiment where we measured the cosine similarity between Adam's first moment and the gradient we compute immediately after we update the target network. If there is no contamination effect then surely the cosine similarity (which is between -1 and +1) should usually be positive.

However, as shown in the attached pdf, we see that while it is true that very early in training the cosine similarity is positive (meaning that moment estimates are not stale), the cosine similarity quickly converges to around zero, indicating that there is no meaningful similarity between Adam's moment estimate and the gradient. This observation manifests itself in all games we tested. This is what we refer to as the contamination effect: without resetting, at each iteration we are in effect poorly initializing Adam, and so the optimizer needs to waste some gradient steps just to unlearn this poor initialization. Resetting is a simple and effective strategy to hedge against this contamination.

Finally, Reviewers aUSe and 59Jp asked us to look at effects of resetting on other standard deep RL algorithms, with DQN as their primary suggestion. In light of their feedback, we ran DQN and IQN with and without resetting and repeated the experiment presented in Figure 4 of the paper. In this case, we benchmarked DQN and IQN for different values of $K$ with and without resetting Adam. We see that a similar result manifests itself in which our resetting algorithms perform better than the standard non-resetting algorithms.

With these new results added, overall we have shown that resetting enhances the performance of 1) Rainbow with Adam, 2) IQN with Adam, 3) DQN with Adam, 4) Rainbow Pro with Adam, 5) Rainbow with RMSProp, and 6) Rainbow with Rectified Adam. We really believe that this is a comprehensive set of results, and that it is reasonable to expect that this phenomenon would generalize across other successors of DQN and Rainbow. As mentioned by reviewer aUSe, this strengthens the significance of our work, so we hope Reviewers aUSe and 59Jp consider these additional experiments in their final evaluation of our paper.

Please take a look at the attached pdf for 1) experiments with added seeds, 2) experiments measuring the contamination effect, and 3) experiments with DQN and IQN.

---

### Decision · Program_Chairs · 2023-09-21

**Decision:**

Accept (poster)

**Comment:**

There were no major concerns for this paper after the rebuttal. The remaining weaknesses are:
* Lowish novelty
* Applicable only to DQN-like algorithms with hard-updates

Overall, the paper is sufficiently interesting and done well-enough to merit acceptance. Given that the proposed change is small and easy to implement, it is likely to have a significant impact. The tangential Rainbow results w/o a target network also quite interesting. I recommend acceptance.